# Pacific subduction control on Asian continental deformation including Tibetan extension and eastward extrusion tectonics

W.P. Schellart [1,2]*, Z. Chen[2,3], V. Strak [1,2], J.C. Duarte [2,4,5] & F.M. Rosas [4,5]

The India-Asia collision has formed the highest mountains on Earth and is thought to account for extensive intraplate deformation in Asia. The prevailing explanation considers the role of the Pacific and Sunda subduction zones as passive during deformation. Here we test the hypothesis that subduction played an active role and present geodynamic experiments of continental deformation that model Indian indentation and active subduction rollback. We show that the synchronous activity and interaction of the collision zone and subduction zones explain Asian deformation, and demonstrate that east-west extension in Tibet, eastward continental extrusion and Asian backarc basin formation are controlled by large-scale Pacific and Sunda slab rollback. The models require 1740 ± 300 km of Indian indentation such that backarc basins form and central East Asian extension conforms estimates. Indentation and rollback produce ~260–360 km of eastward extrusion and large-scale clockwise upper mantle circulation from Tibet towards East Asia and back to India.

---

[1] Department of Earth Sciences, Vrije Universiteit Amsterdam, Amsterdam, Netherlands. [2] School of Earth, Atmosphere and Environment, Monash University, Melbourne, VIC 3800, Australia. [3] Japan Agency for Marine-Earth Science and Technology, Yokohama, Japan. [4] Instituto Dom Luiz (IDL), Faculdade de Ciências, Universidade de Lisboa, Lisboa, Portugal. [5] Departamento de Geologia, Faculdade de Ciências, Universidade de Lisboa, Lisboa, Portugal. *email: w.p.schellart@vu.nl

The Eurasian plate has accommodated significant short-ening due to India–Asia convergence since the Early Cenozoic onset of collision, with estimates in the range 1000–2000 km[1–3]. Such estimates, however, fall significantly short to explain the total amount of convergence of ~2400–3600 km[4–7]. This could be because convergence is partly accommodated through continental subduction[8], because of uncertainty in east-ward extrusion estimates, ranging from ~250 to 1250 km[5,9,10], because proposed shortening values underestimate actual short-ening in the intraplate deformation zone or a combination of the above. In the current paradigm for the Cenozoic evolution of this intraplate zone, all deformation, including shortening, strike-slip and extensional deformation (Fig. 1a), is ascribed to the collision and progressive indentation of the Indian subcontinent into the Eurasian plate[1,9,11,12]. Moreover, backarc basin formation in the Japan Sea, Kuril Basin and the Sea of Okhotsk (Fig. 1a) has been interpreted as a far-field effect of the India–Eurasia collision[13–15].

Although India–Eurasia convergence has driven Himalayan mountain building and Tibetan Plateau formation, its role in extensional faulting along north–south striking grabens in Tibet and in widespread deformation outside the zones of mountain building remains speculative and has been questioned. Early works have argued in favour of a role for the Pacific margin in driving extension in East China specifically[16,17], and East Asia more generally[18]. In the past two decades, conceptual models have been proposed that argue for an active role of the Pacific subduction margin in driving East Asian extension through slab rollback[10,19,20]. Extension along the East and Southeast Asian margins suggests an active role for the Pacific and Sunda sub-duction zones, in particular considering the continuity of extension in space and time (Fig. 1a). Indeed, extensional deformation and strike-slip faulting are observed over an enor-mous area covering some 30 million square kilometres, from Indonesia to Kamchatka. This deformation took place across the entire region during the Eocene–Oligocene. In several locations, extension was also active in the latest Cretaceous/Paleocene (e.g. Beibuwan basin, Pearl River Mouth basin, Taiwan Strait basins, East China Sea basins, Yellow Sea basins, Bohai basin, Hei-longjiang basin), thus preceding collision[18], and/or during part of the Miocene–Present (e.g. Mergui basin, Banda Sea, southern South China Sea, Okinawa Trough, Japan Sea and margins, Kuril Basin).

Previous geodynamic models simulating Asian deformation used an active rigid indenter to simulate India progressively moving into and colliding with the Asian lithosphere, thereby applying compressive deviatoric normal stresses[9,12,19]. Another model focused on gravitational spreading of the East Asian lithosphere and did not include an active Indian indenter[20]. All these earlier models used passive lateral boundaries in East and Southeast Asia with zero stress or lithostatic stress boundaries, simulating passive subduction zone rollback (i.e. subduction zone migration that resulted only from the Indian indenter and/or gravitational spreading and not from forces originating from the subduction zone itself). However, geodynamic models show that subduction zones actively deform overriding plates during slab rollback by applying deviatoric stresses at the subduction zone interface and flow-induced shear tractions at the base of the overriding plate[21–23].

Recent studies have investigated the formation of the Himalaya and the Tibetan Plateau through modelling subduction and continental subduction[24,25]. Such models involved a relatively limited spatial domain of Asian continental lithosphere and excluded the Western Pacific subduction margin and most of the Sunda subduction margin. Our current study focusses on a much larger domain of Asian continental lithosphere (about an order of magnitude larger), stretching from the Himalaya and Tibet in the west, Indonesia in the southeast and Kamchatka in the northeast (Fig. 1). Here we investigate the role of the Western Pacific subduction margin and the entire Sunda subduction margin in Asian deformation and their interaction with Indian indenter tectonics and large-scale mantle flow. Our models reveal that the synchronous activity and interaction of the collision zone and subduction zones are crucial for explaining the entire deforma-tion field in Central, East and Southeast Asia and demonstrate that enigmatic east–west extension in Tibet, eastward continental extrusion and backarc basin formation along the East and Southeast Asian margins are controlled by large-scale Pacific and Sunda slab rollback. The experiments constrain the amount of Indian indentation, thereby predicting the amount of eastward continental extrusion. Our quantification of Indian indentation and Western Pacific rollback also makes predictions on the large-scale mantle flow in the region and implies large-scale clockwise upper mantle circulation from the Tibetan region towards East Asia and from the Philippine Sea region along a path south of the Sunda–Banda slab wall into the Indian Ocean domain. As such, our experiments demonstrate the crucial role that the Western Pacific and Sunda subduction zones have played in actively deforming the continental lithosphere and underlying mantle in Central, East and Southeast Asia.

## Results

**Experimental approach.** Here we present the first geodynamic models of widespread continental deformation in Central, East and Southeast Asia combining two separate approaches that simultaneously simulate active Indian indentation[12] and active rollback[26] of the Western Pacific and Sunda subduction zones (Fig. 2). The active rollback boundaries apply deviatoric tensile stress conditions (trench suction) and slab rollback-induced basal mantle flow tractions to the overriding continental Eurasian lithosphere. The analogue experiments use glass microspheres with a frictional plastic rheology to simulate the brittle upper continental lithosphere and high-viscosity silicone to simulate the viscous continental lower lithosphere of Eurasia. Low-viscosity glucose syrup is used to simulate the low-viscosity sub-litho-spheric mantle for isostatic compensation and mantle flow. The models are scaled for gravity, including gravitational body forces and potential energy differences between the continental and oceanic domains. Furthermore, we implement length scaling and geometrical aspect ratios such that, for the first time, the model components accurately represent the size of the Indian indenter (~2400 km), the western Pacific subduction margin (~8000 km), the Sunda subduction zone (~4000 km) and the Asian litho-spheric thickness (~104 km) at the onset of collision (see "Methods").

We test the role of the advance rate (indentation rate) of the India–Eurasia convergent boundary, i.e. the Indian continental subduction zone hinge and slab ($v_I$), relative to the rollback rate of the Western Pacific ($v_{WP}$) and Sunda ($v_{Su}$) subduction zones on the style and extent of continental deformation in Asia. In three experiments, we test different rates for $v_I$ (keeping $v_{WP}$ and $v_{Su}$ constant). We test minimum and intermediate rates of 2.0 cm year$^{-1}$ (experiment $I_{MIN}$-R) and 3.6 cm year$^{-1}$ (experiment $I_{INT}$-R) based on minimum and maximum Asian shortening estimates of ~1000 and 1800–2000 km, respectively[2,3], averaged over the past ~52 Myr. And we test a maximum rate of 5.2 cm year$^{-1}$ (experiment $I_{MAX}$-R) assuming that the India–Eurasia conver-gence rate averaged over ~52 Myr[6] is accommodated entirely by shortening in Eurasia ("Methods"). In two additional experi-ments, we test Indian indentation without Pacific–Sunda rollback ($v_{WP} = v_{Su} = 0$) (experiment $I_{INT}$-NR) and Pacific–Sunda roll-back without Indian indentation ($v_I = 0$) (experiment NI-R).

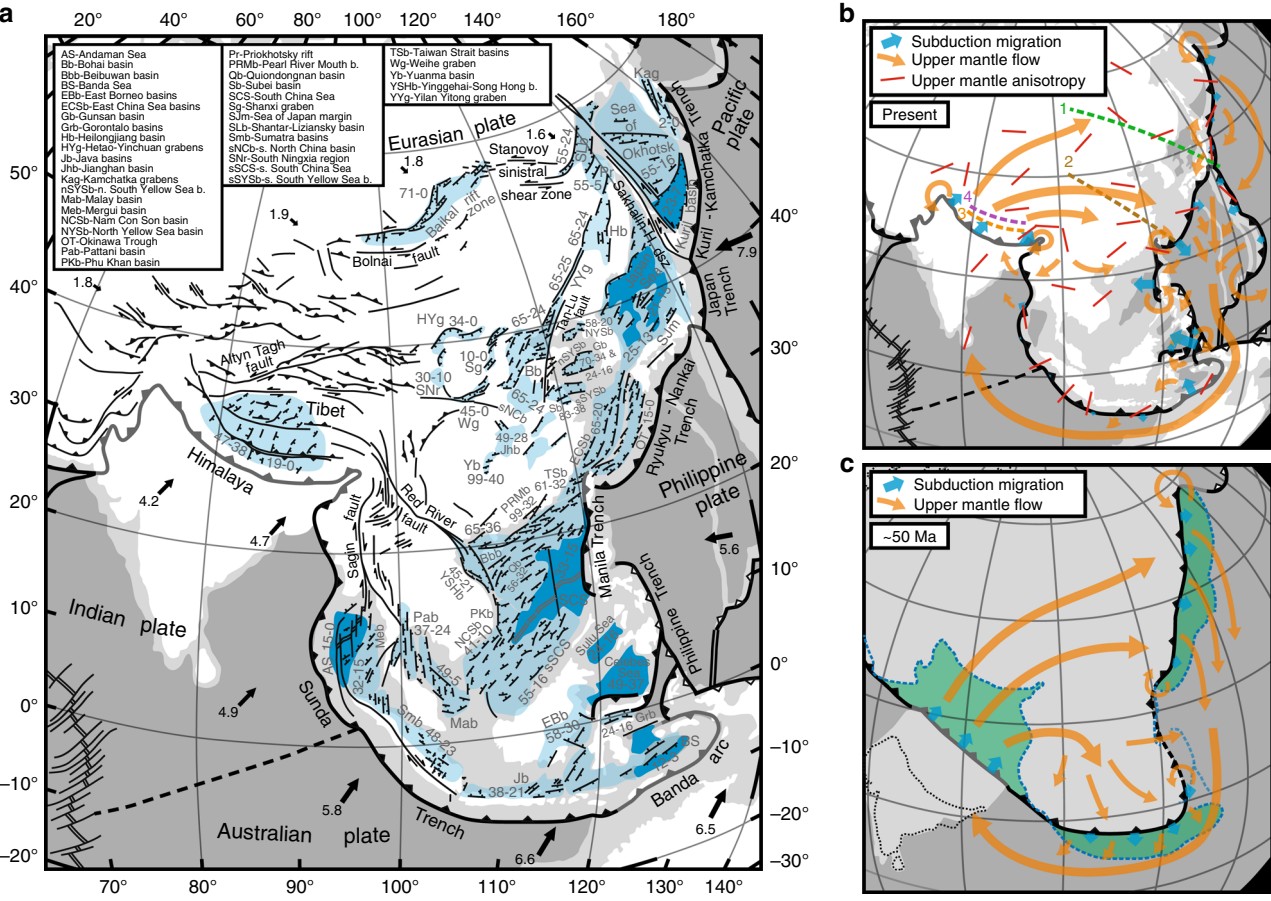

**Fig. 1** Structural-tectonic maps showing widespread Cenozoic continental deformation in Central, East and Southeast Asia. **a** Structural-tectonic map showing structures and ages of extensional basins (see Supplementary Note 1). Arrows indicate present-day plate velocities (in cm year$^{-1}$) in the Indo-Atlantic hotspot reference frame[81]. Numbers next to extensional structures/basins indicate timing of extensional activity (in Ma). Light blue areas represent extended continental crust, dark blue areas represent backarc/marginal basin oceanic crust. **b**, **c** Present-day tectonic setting and simplified tectonic reconstruction at ~50 Ma of Central, East and Southeast Asian convergent plate boundaries showing oceanic/continental subduction zone migration (blue arrows) and proposed large-scale upper mantle flow patterns (orange arrows) that accommodate lateral slab migration (Indian continental slab advance and predominant retreat for oceanic slabs). Red bars in **b** indicate regional pattern of upper mantle anisotropy based on seismic shear wave splitting measurements derived from various sources[45,46,49,82]. Dashed lines with numbers 1–4 in **b** indicate sections along which Cenozoic finite extension has been estimated (see "Methods"). Blue dashed lines in **c** indicate present-day position of main Asian convergent boundaries to illustrate finite plate boundary migration since ~50 Ma. Green areas in **c** illustrate surface areas of upper mantle volumes displaced since ~50 Ma by lateral slab migration. The simplified reconstruction in **c** is based on earlier reconstructions for the Himalaya–Tibet region and Southeast Asia[74,75], for the East China region[17], for the Japan region[76,77] and for the Kuril-Kamchatka-Sea of Okhotsk region[26]

Below we show that the style and distribution of Asian deformation depends critically on the relative velocities of $v_I$ and $v_{WP}$, expressed as ratio $R = (v_I - v_{WP})/(v_I + v_{WP})$, and on the amount of Indian indentation ($I_I$).

**Experimental results of Asian deformation.** The experiments all show the development of a fold-and-thrust belt north of the Indian indenter, comparable to nature and earlier models[12,19], which becomes increasingly asymmetric with decreasing $R$ (Figs. 3–5, Supplementary Figs. 1–4). By increasing Indian finite indentation from $I_{MIN}$-R to $I_{INT}$-R to $I_{MAX}$-R, extension in East Asia is progressively suppressed, in particular in the central East Asian region located west of the Japan, Ryukyu and Manila subduction zones, but strike-slip faulting and eastward extrusion are enhanced (Figs. 3a–c, 4a–c, 6a, b). Strike-slip faulting is mostly confined to the regions east and west of the indenter and to a zone extending from the northeast corner of the indenter to the Kuril–Kamchatka region, which accommodates sinistral shear. The deformed grid and displacement field show eastward

and southeastward extrusion of continental material but only with active Pacific and Sunda rollback (Figs. 3, 5 and 6b, Supplementary Fig. 3). The infinitesimal displacement fields for the last stage of the experiments (Fig. 5a–c), which represent the present-day displacement field, are generally consistent with and comparable to the Global Positioning System (GPS) velocity field observed in Central and East Asia[27]. Indeed, both experimental and observational fields show approximately radially divergent vectors north of the Indian indenter that decrease in length with increasing distance from the indenter, east-directed vectors in central and northern East Asia and a clockwise rotating vector pattern near the northeast corner of the indenter and in Southeast Asia. For models $I_{MIN}$-R and $I_{INT}$-R, the east-directed vectors in central and northern East Asia increase in length towards the east (Fig. 5a, b) but they decrease for model $I_{MAX}$-R (Fig. 5c).

With low $I_I = $ ~1049 km and $R = 0.25$, strike-slip faulting is limited but extensional deformation in East and Southeast Asia is penetrative, with normal faulting and rifting distributed over a wide area (Figs. 3a and 4a). With intermediate indentation ($I_I = $

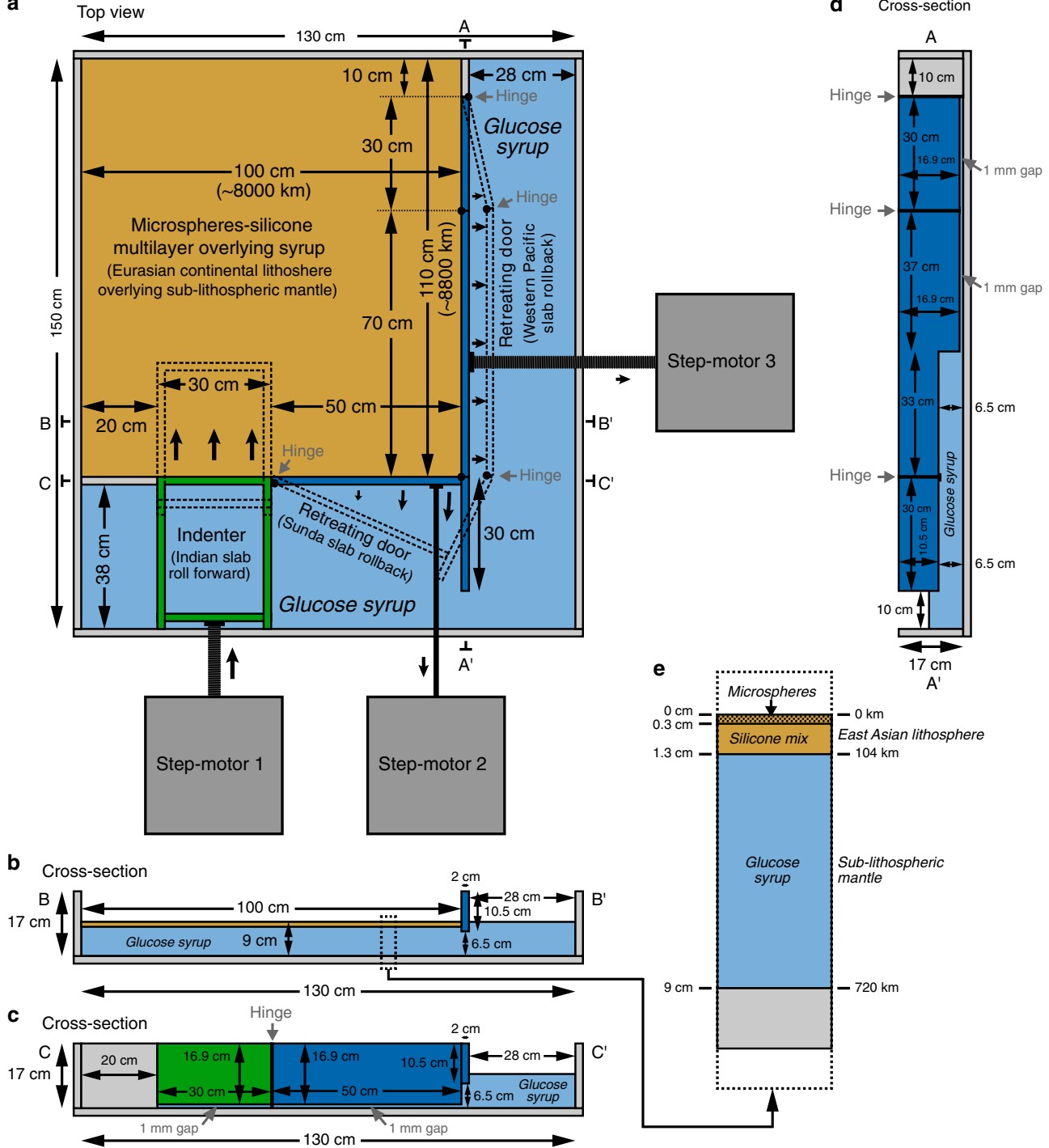

**Fig. 2** Schematic diagrams illustrating the experimental apparatus set-up. The experiments investigate widespread deformation of a layered brittle–ductile system simulating part of the Eurasian continental lithosphere. **a** Top view of the experimental apparatus. Deformation in the model lithosphere is induced by northward motion of a rigid boundary (simulating progressive indentation of India into Eurasia and roll forward of the Indian slab and hinge) and outward motion of two retreating boundaries (simulating eastward slab rollback of the Western Pacific subduction margin and southward rollback of the Sunda slab). Motion of the three boundaries is driven by three step-motors. **b**, **c** East–west cross-sectional views through the model lithosphere and the southern boundary. **d** Cross-sectional view through the eastern boundary. **e** Vertical profile of the model layers with a frictional-plastic top layer made of fine-grained glass microspheres to simulate the brittle upper lithosphere, a linear-viscous layer of filled silicone oil simulating the ductile lower lithosphere (dynamic shear viscosity of $5.8 \pm 0.2 \times 10^4$ Pa s), and a bottom layer made of low-viscosity glucose syrup with a linear viscous rheology (dynamic shear viscosity of $254 \pm 7$ Pa s) to simulate the sub-lithospheric mantle. Note that only the inner compartment of the box is filled with the layered system (continental lithosphere), while the entire box is filled with the low-viscosity bottom layer (sub-lithospheric upper mantle) to isostatically support the model continental lithosphere. Also note that 1 cm in the model represents 80 km in nature

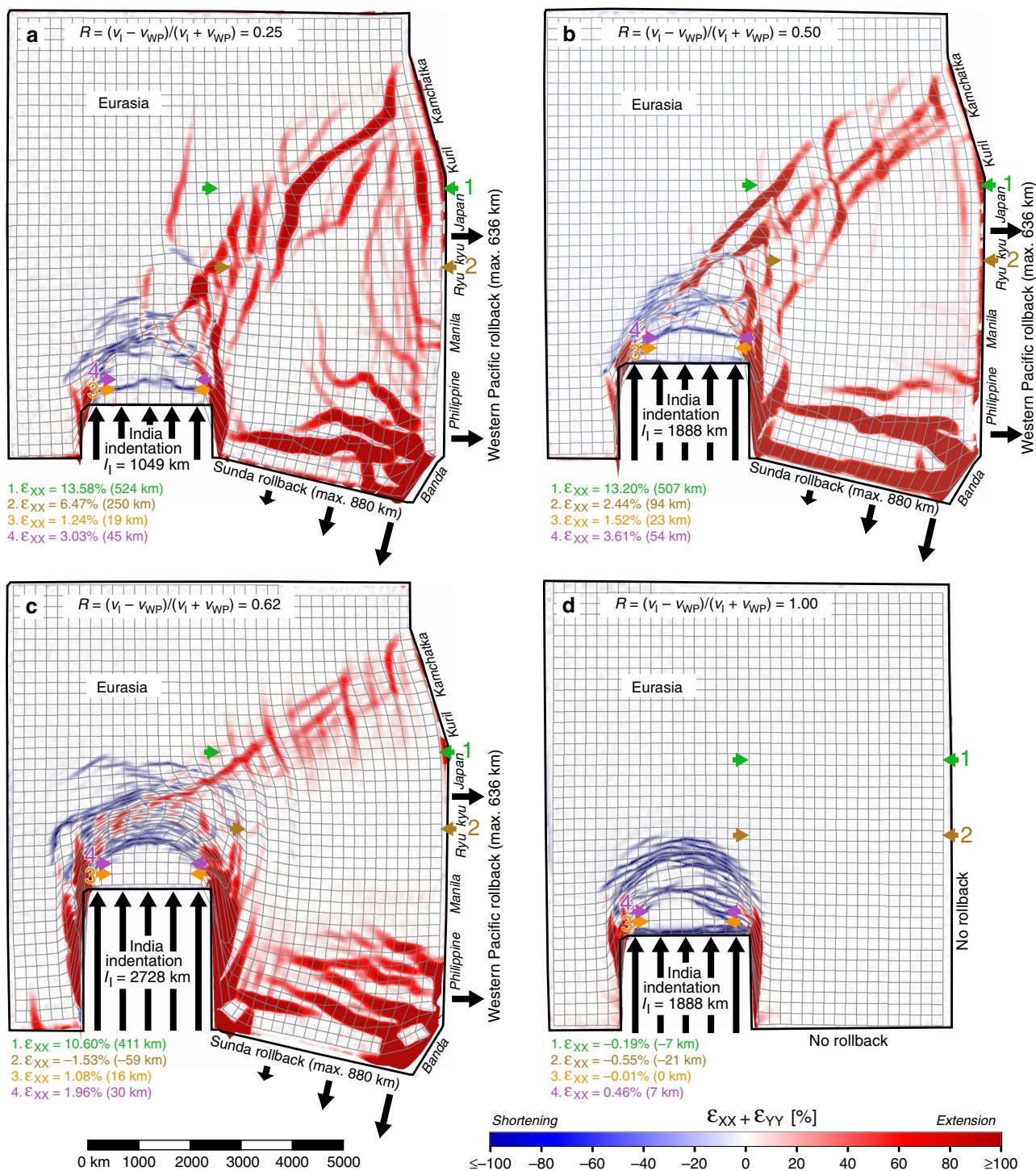

**Fig. 3** Top views of four experiments with different velocity ratios showing finite horizontal normal strain ($\varepsilon_{XX} + \varepsilon_{YY}$) and deformed model grid. The experiments simulate Asian continental deformation and the velocity ratio is expressed as $R = (v_I - v_{WP})/(v_I + v_{WP})$, where $v_I =$ Indian continental subduction hinge and slab advance (roll-forward) velocity and $v_{WP} =$ Pacific subduction hinge and slab retreat (rollback) velocity. The results are shown for the end of each experimental run. **a** Experiment $I_{MIN}$-R with $R = 0.25$ (minimum indentation). **b** Experiment $I_{INT}$-R with $R = 0.50$ (intermediate indentation). **c** Experiment $I_{MAX}$-R with $R = 0.62$ (maximum indentation). **d** Experiment $I_{INT}$-NR with $R = 1.00$ (intermediate indentation and no rollback). Note that the colour scheme indicates $\varepsilon_{XX} + \varepsilon_{YY}$. The locations of four east–west sections (1–4) along which east–west finite strain ($\varepsilon_{XX}$) and finite extension are calculated are indicated with green, brown, orange and purple arrows (extension is positive, shortening is negative). These sections are the model equivalents of sections 1–4 shown in Fig. 1b. Experimental results showing the horizontal finite strain ellipse field, digital photographs, horizontal finite displacement field and surface topography are presented in Supplementary Figs. 1–4. Four evolutionary stages of experiments $I_{INT}$-R and $I_{MIN}$-R are shown in Supplementary Figs. 5 and 6, respectively

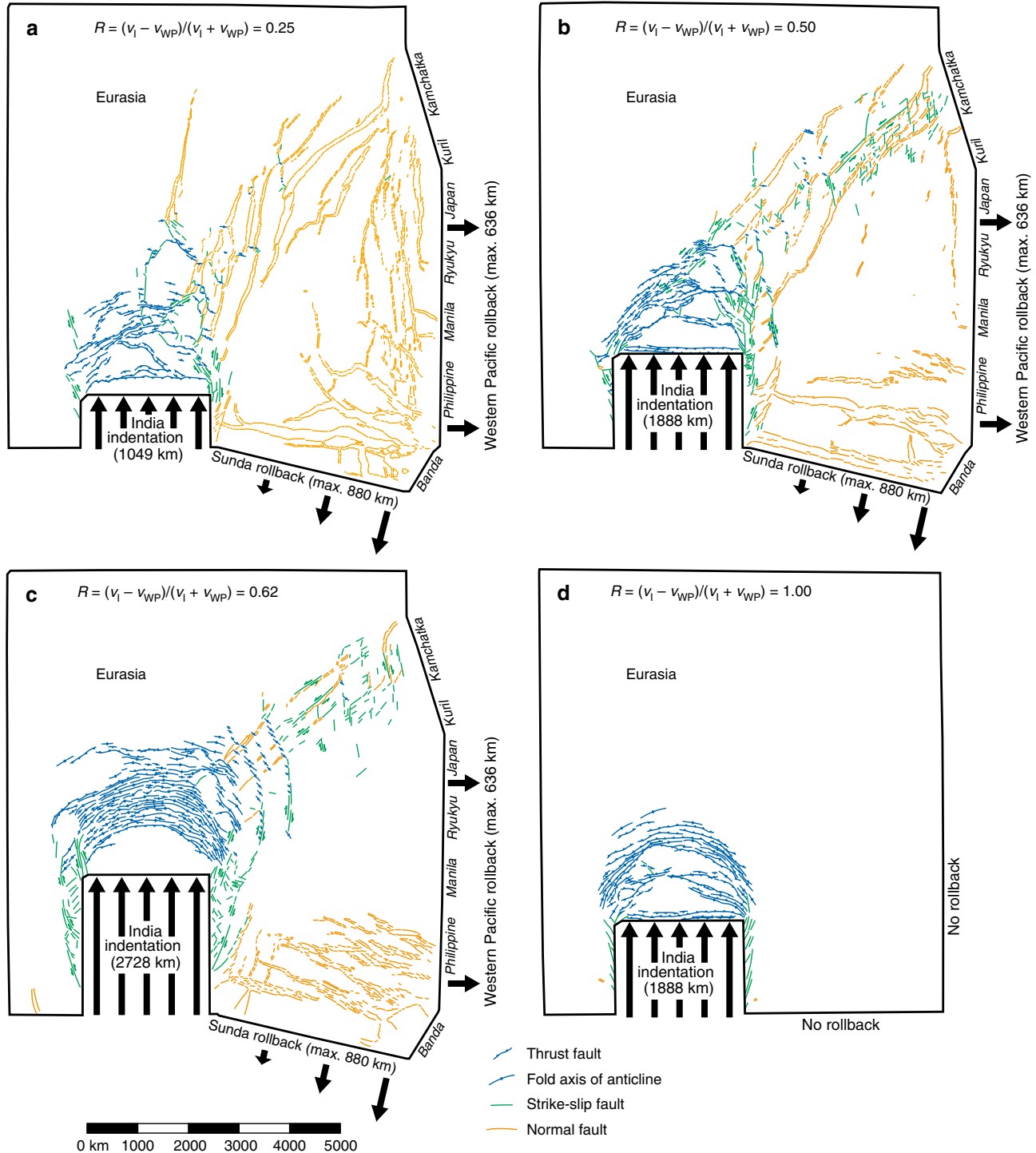

**Fig. 4** Top views of four experiments simulating Asian deformation with different velocity ratios ($R$) showing the surface structures. The results are shown for the end of each experimental run. **a** Experiment $I_{MIN}$-R with $R = 0.25$ (minimum indentation). **b** Experiment $I_{INT}$-R with $R = 0.50$ (intermediate indentation). **c** Experiment $I_{MAX}$-R with $R = 0.62$ (maximum indentation). **d** Experiment $I_{INT}$-NR with $R = 1.00$ (intermediate indentation and no rollback). Note that thrust faults and anticlinal fold axes are in blue, strike-slip faults are in green and normal faults are in orange

~1888 km) and $R = 0.50$, normal faults, rifts and grabens are still widely distributed but smaller in number and size, while strike-slip faults are more pronounced (Figs. 3b and 4b), a number of which form conjugate pairs. Interestingly, model $I_{INT}$-R shows an asymmetry of strike-slip faulting east and west of the Indian indenter with significant strike-slip faulting along its eastern

boundary, dominated by dextral north–south to NNE–SSW striking faults and lesser conjugate ENE–WSW striking sinistral faults but limited strike-slip faulting along its western boundary (Fig. 4b). This is in agreement with the first-order strike-slip faulting patterns observed east and west of the Indian indenter, showing limited strike-slip faulting west of India and significant

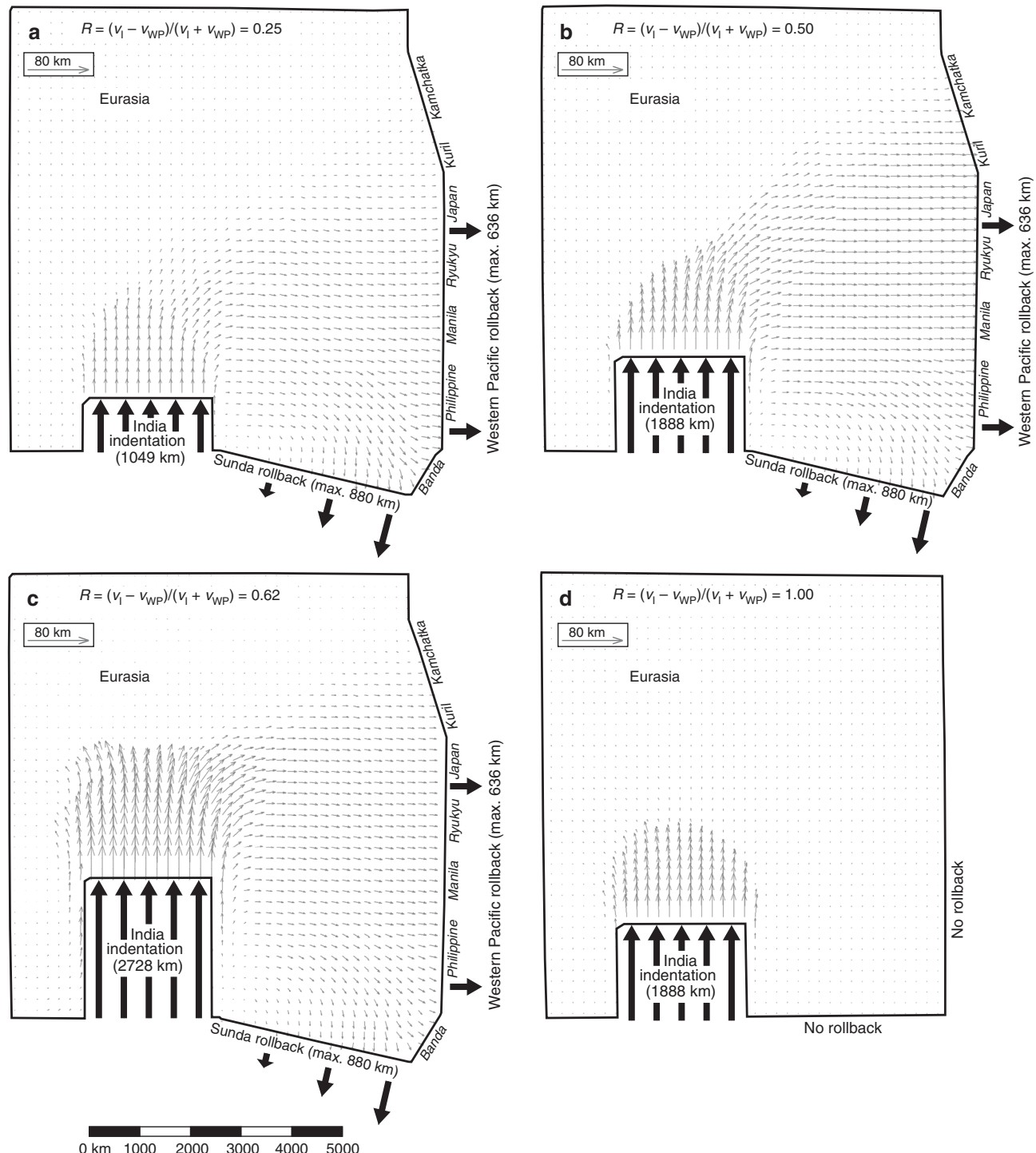

**Fig. 5** Top views of four experiments simulating Asian deformation with different velocity ratios ($R$) showing the horizontal infinitesimal displacement field. The results are shown for the end of each experimental run. **a** Experiment $I_{MIN}$-R with $R = 0.25$ (minimum indentation). **b** Experiment $I_{INT}$-R with $R = 0.50$ (intermediate indentation). **c** Experiment $I_{MAX}$-R with $R = 0.62$ (maximum indentation). **d** Experiment $I_{INT}$-NR with $R = 1.00$ (intermediate indentation and no rollback). The infinitesimal displacement field is determined for the last 40 min (~0.866 Myr) of the experiment. Note that the displacement fields in **a**–**c** generally mimic the GPS velocity field observed in Central, East and Southeast Asia[27]

strike-slip faulting east of India with north–south striking dextral faults (e.g. Sagin fault) and lesser conjugate ENE–WSW striking sinistral faults (Fig. 1a).

The widespread extensional structures as observed in $I_{MIN}$-R and $I_{INT}$-R are absent in earlier experimental models that lack active Pacific and Sunda rollback[9,12,19]. The extensional structures in $I_{MIN}$-R and $I_{INT}$-R occur up to ~5000 km (for $I_I = $ ~1049 km)

and ~4000 km (for $I_I = $ ~1888 km) from the Pacific subduction boundary (Fig. 4a, b), comparable in distance to the far-field extension in the Baikal rift zone.

Along the Pacific subduction boundary, the nearest rift and graben structures form some 400–600 km west of this boundary (Fig. 4a, b), which is comparable to the distance between the Pacific trench and the Okinawa, Japan, Okhotsk and Kuril

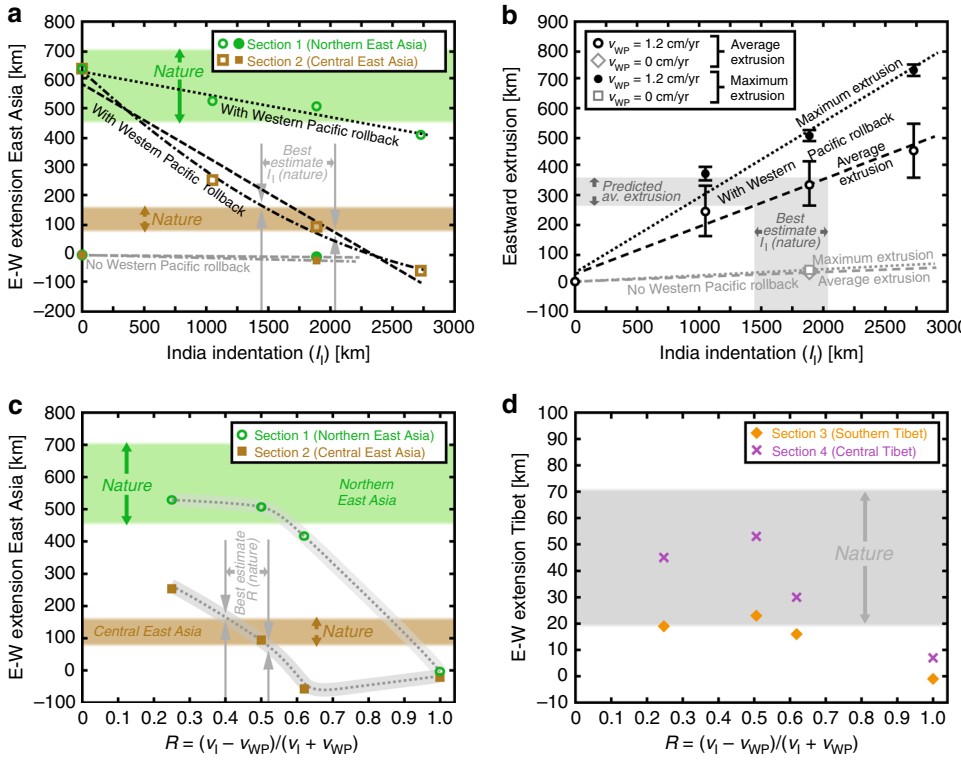

**Fig. 6** Continental extension and extrusion as a function of Indian indentation ($I_I$) and indenter-rollback ratio R. **a** East–west extension as a function of $I_I$ along two sections (1 and 2) for experiments with Western Pacific slab rollback (open symbols) and without Western Pacific slab rollback (filled symbols) and for the representative sections in nature (see "Methods"). **b** Average and maximum eastward continental extrusion as a function of $I_I$ for experiments with Western Pacific slab rollback (circles) and without Western Pacific slab rollback (diamond and square). Predicted indentation and predicted average eastward extrusion in nature are shown in **a** and **b**, respectively. Black dashed and black dotted lines in **a** and **b** are least-squares linear best-fit lines (coefficients of determination of 0.964 (**a**, dashed), 0.951 (**a**, dotted), 0.970 (**b**, dashed) and 0.980 (**b**, dotted)). Correlations are statistically significant at 95% (**a**, dotted, p value = 0.028), 98% (**a**, dashed, p value = 0.019), 99% (**b**, dotted, p value = 0.008) and 98% (**b**, dashed, p value = 0.014) confidence level using Fisher's z. Black dash-dot line in **a** is quadratic best-fit line (coefficient of determination 0.996). **c**, **d** Finite east–west extension in East Asia and Tibet as a function of R along experimental sections 1 and 2 and sections 3 and 4, respectively, and the ranges for these sections in nature (see "Methods"). For section locations in nature and experiments, see Figs. 1b and 3, respectively. Note that errors in measuring experimental values are smaller than the sizes of the individual markers. Also note that the black bars in **b** are not error bars but represent the minimum and maximum values measured for the average eastward extrusion and the maximum eastward extrusion as determined from the grid lines in Fig. 3 originally located north of the easternmost 600–850 km of the Indian indenter

backarc basins. The extensional structures mostly run sub-parallel to the Pacific and Sunda subduction boundaries, both in the experiments (Fig. 4a, b) and in nature (Fig. 1a). However, they also strike obliquely to perpendicularly to the boundaries in some regions (both in the experiments and in nature), such as in the northeastern region (Sea of Okhotsk), which results from the asymmetric rotational rollback of the Pacific slab[26], and in the southeast, which results from the interference pattern of deformation produced by the two retreating subduction zones, in combination with asymmetric rollback. The occurrence of obliquely oriented extensional structures, postulated oblique backarc spreading ridges, and their spatial association with strike-slip faults in the Japan–Kuril–Okhotsk domain has been used as an argument against slab rollback as their driving agent. The current model results and an earlier regional study[26] indicate that such obliquely oriented extensional structures and strike-slip faults can in fact result directly from subduction zone processes along the Sunda and Western Pacific plate boundaries.

In case of a high $I_I = {\sim}2728$ km and $R = 0.62$ (experiment $I_{MAX}$-R), folds, thrusts and strike-slip faults, the latter often in conjugate pairs, are very pronounced, while normal faulting is limited to the southeast and northeast (Figs. 3c and 4c). Notably, normal faulting and rifting are absent in central East Asia (west of

Japan, Ryukyu and Manila), resulting in a ${\sim}3000 \times 3000$ km² area that lacks east–west extension but instead shows east–west shortening with approximately north–south striking zones of dextral transpression in the western part of the area (Fig. 4c). Also, no extensional structures form along the Western Pacific subduction boundary, in disagreement with observations of Cenozoic backarc extension along the Pacific margin and extension in East China (Fig. 1a). In addition, model $I_{MAX}$-R shows a symmetrical pattern of strike-slip faulting east and west of India (Fig. 4c), which is not in agreement with nature with significant strike-slip faulting along the eastern boundary but limited strike-slip faulting along the western boundary (Fig. 1a).

In the experiment that lacks Western Pacific and Sunda rollback ($I_{INT}$-NR, $R = 1$), eastward extrusion is negligible (Figs. 3d, 5d and 6b, Supplementary Fig. 3d) and the regions east and northeast of the collision zone show minor east–west shortening (Figs. 3d and 6c), in disagreement with observations. Furthermore, no deformation structures are observed in East and Southeast Asia (Fig. 4d), which is also in disagreement with observations (Fig. 1a).

Four evolutionary stages of models $I_{INT}$-R and $I_{MIN}$-R are shown in Supplementary Figs. 5 and 6, respectively, illustrating that shortening north of the Indian indenter and extension north

and west of the Sunda and Western Pacific subduction margins, respectively, are active from the earliest stages. Such early extension is consistent with extension observed in East and Southeast Asia that starts already in the Early Cenozoic, such as the Sumatra, Java and East Borneo Basins, the margins of the South China Sea, the basins in East and Northeast China, the Baikal Rift zone and the basins in the Sea of Okhotsk region (Fig. 1a).

**Pacific subduction drives east–west extension in Tibet.** Tibet is characterized by extensional structures, which are best developed in southern and central Tibet at 200–800 km from the India–Eurasia plate boundary. The extensional structures consist of north to north–northeast striking normal faults, grabens and dikes[28] active since at least ~19 Ma[7,29], with some dikes with reported activity at 47–38 Ma[30]. The extensional structures have been ascribed to various driving mechanisms that have been simulated in geodynamic models focussing on the Himalaya–Tibet region, including gravitational collapse of the plateau that is triggered by convective removal of the lithospheric root[11], underthrusting of India and related basal shear tractions[31] and lower crustal flow[32]. Our geodynamic models focus on a much larger spatial domain, about an order of magnitude larger, than these earlier models and do not incorporate convective removal of the lithospheric root, nor Indian underthrusting nor lower crustal flow. Yet, our models reproduce east–west extension in the Tibetan Plateau region (Figs. 3 and 6), indicating that the earlier proposed mechanisms are not essential for reproducing the east–west extension. As such, our models provide an alternative driving mechanism in which east-directed rollback of the

Western Pacific subduction boundary and associated mantle flow (Fig. 7) induce east–west extension north of the Indian indenter. Several tens of kilometres of east–west extension in southern and central Tibet are reproduced in our experiments but only for those experiments that have eastward Pacific slab rollback ($I_{MIN}$-R, $I_{INT}$-R and $I_{MAX}$-R, Figs. 3 and 6d). These experiments can also account for a possible Eocene onset of east–west extension[30], as Pacific rollback was already active during this time. There is no east–west extension in southern Tibet without Pacific rollback ($I_{INT}$-NR with $R = 1$; Fig. 6d), suggesting that such rollback is indeed required for extension to take place. Estimates of total east–west extension in Tibet range from 20 to 70 km[2,28,33], comparable to our experiments with $R = 0.25$–0.62 showing 16–54 km of east–west extension.

Our general finding that the Pacific subduction zones can affect the continental deep interior of Asia, including Tibet and the Baikal region located many thousands of kilometres (~3000–4000 km) from the subduction margin, has implications for the extension recorded in ancient orogens. Examples include the Paleozoic Variscan, Caledonian and Appelachian orogens, which experienced syn-orogenic, late-orogenic and/or post-orogenic extension that could have resulted from far field subduction forces in a manner alike that for East Asia. Indeed, subduction zones were located far from these mountain belts at the time of extension.

**Constraining Indian indentation.** The least-squares best-fit trend lines for experimental data showing central East Asian extension as a function of Indian indentation $I_I$ (Fig. 6a) can be used to constrain $I_I$ in nature through comparison with observed

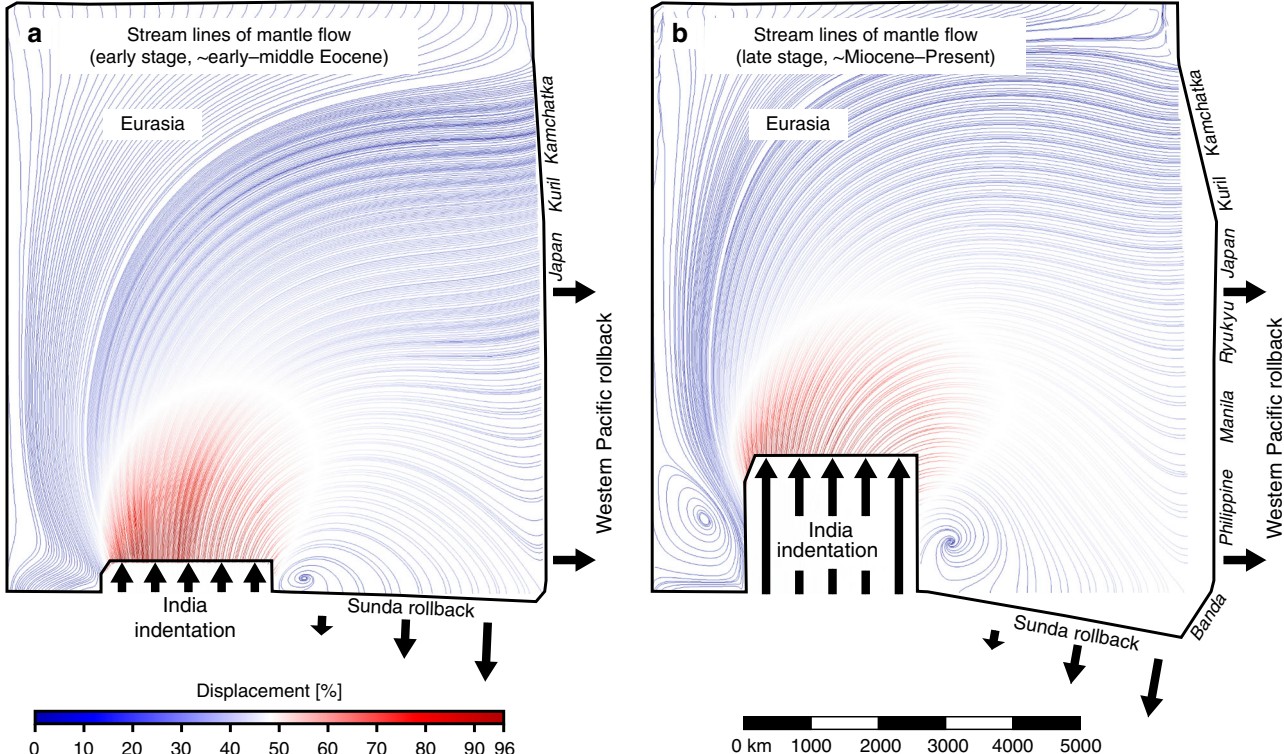

**Fig. 7** Top views of experiment I-R$_{MO}$ showing the horizontal mantle streamline pattern. The streamline pattern illustrates the general flow pattern at the lithosphere–asthenosphere boundary and in the ductile lithospheric mantle and results from Indian indentation (continental subduction hinge and slab advance) and Western Pacific and Sunda subduction zone rollback (hinge and slab retreat). **a** Horizontal mantle flow pattern for an early stage of the experiment (corresponding to the early–middle Eocene). **b** Horizontal mantle flow pattern for a late stage of the experiment (corresponding to the Miocene–Present). Note that I-R$_{MO}$ has an $R = 0.62$ and no brittle top layer

extension. The best-fit lines imply that $I_I \leq \sim 2040$ km, because otherwise the trend lines predict too little extension (<75 km) or east–west shortening in central East Asia, which is in disagreement with geological observations of 75–165 km of extension. Additionally, $I_I \geq \sim 1440$ km, because otherwise the trend lines predict extension in central East Asia that is larger than observed. The above leads us to conclude that $1740 \pm 300$ km of India–Eurasia convergence since the start of collision has been accommodated by indentation (Indian subduction hinge advance) causing Central, East and Southeast Asian deformation, in agreement with recent data[3]. Such indentation predicts ~260–360 km of average eastward extrusion (Fig. 6b), comparable to estimates of ~250 km based on tectonic reconstructions[5] but significantly less than high values (800–1250 km) proposed earlier[9,10]. Indian indentation amounts to about one-half to two-thirds of the total estimated convergence, the remainder of which has to have been accommodated by Indian continental subduction, for which there is evidence[8,34].

## Discussion

The first-order agreement between nature and our best fitting model $I_{INT}$-R (and to a lesser extent model $I_{MIN}$-R) is striking and relates to the spatial distribution and orientation of thrust faults, strike-slip faults and normal faults, to the morphology of the mountain ranges, to the amount of extension in East Asia, Southeast Asia and Tibet and to geodetic observations. Regarding shortening and mountain building, the main agreements include shortening structures north of the Indian indenter with approximately east–west to NNE–SSW striking folds and thrust faults (Figs. 1a, 3a, b and 4a, b), a mountainous region north of the Indian indenter that is narrower in the west and widens eastward, similar to the Himalaya–Tibet mountains (Supplementary Fig. 4), and topographic elevations of the experimental mountain ranges in the Himalaya–Tibet region with respect to the undeformed foreland that scale to 5.3–9.2 km ($I_{INT}$-R) and 3.5–6.7 km ($I_{MIN}$-R) in nature (Supplementary Fig. 4). As for strike-slip faulting, the main similarities are dextral, approximately north-south striking, strike-slip faults east of the indenter and associated conjugate sinistral faults, similar to the Sagin fault and conjugates east of India, and less developed sinistral, approximately north-south striking strike slip faults west of the indenter (Figs. 1a, 3a, b and 4a, b), as well as sinistral, NE–SW striking strike-slip faults that run from the northern edge of the mountainous region to the northeast (Sea of Okhotsk region), similar to sinistral shear zones such as the Altyn Tagh fault, Bolnai fault and Stanovoy sinistral shear zone (Figs. 1a, 3b and 4b). The agreement relating to normal faulting and extension includes normal faults, rifts and grabens in East Asia that strike predominantly sub-parallel to the strike of the Western Pacific subduction margin and occur up to several thousand km from the margin (Figs. 1a and 4a, b), normal faults, rifts and grabens near the Sunda margin that strike predominantly sub-parallel to the strike of this subduction margin (Figs. 1a and 4a, b) and a subordinate number of normal faults, rifts and grabens in East and Southeast Asia that strike at an oblique angle or sub-perpendicularly to the Western Pacific and Sunda subduction margins (Figs. 1a and 4a, b). We note that there is also agreement in terms of extension magnitude, with major extension in northern East Asia, moderate extension in central East Asia and minor east–west extension in the Tibetan region (Fig. 6a–d). Finally, there is also agreement between the best-fitting models and nature in terms of displacements, with east- and southeast-directed displacement fields in East and Southeast Asia, respectively, that are reproduced in the experiments (Fig. 5a, b).

Some discrepancies between observations and models are also evident, such as the occurrence in nature of local zones of compressive tectonics along the Pacific and Sunda margins during the latest Cenozoic. For example, a local segment of the Sunda convergent margin in the southeasternmost part of Southeast Asia (Timor–Banda segment) has experienced shortening tectonics due to continental subduction of Australian continental passive margin, which initiated ~3.5 Ma[35], while backarc spreading and extensional tectonics in the Banda Sea took place until as recently as 3 Ma[36] (Fig. 1a). Along the Pacific margin, Taiwan and northern Honshu Island in Japan have also experienced shortening, but again, these are relatively local phenomena compared to the scale of East and Southeast Asia and they have only been active since 3–2 Ma for Taiwan[37] and ~3.5 Ma for northern Honshu[38]. There are also reports of short-lived (~1–2 Myr) inversion and compression for older times along the East and Southeast Asian margins, such as during the late Oligocene in the southern Sumatra basin[39] and the Chezhen basin, a sub-basin of the Bohai basin in Northeast China[40]. Such local, short-lived phases of overriding plate compression and shortening can be explained, for example, by subduction of a short aseismic ridge or small oceanic plateau[41]. These phases of compression, shortening and inversion are thus local and lasted only for a short duration, while the overall, large-scale, tectonics of East and Southeast Asia during most of the Cenozoic has been dominated by extension, as is evident from Fig. 1a.

Another discrepancy between our models and observations relates to large dextral strike-slip faults located near, and running sub-parallel to, or striking obliquely to, the East Asian margin, such as the Tan–Lu fault and the Sakhalin–Hokkaido dextral shear zone (Fig. 1a). The latter, bordering the Sea of Okhotsk–Kuril basin backarc domain, has up to several hundred kilometres of dextral offset and has been explained by local asymmetrical slab rollback of the Kuril–Kamchatka subduction segment[26]. The Tan–Lu fault, although hundreds of kilometres long, has a Cenozoic dextral offset of only ~21 km[42]. Furthermore, this fault and the Bohai Bay basin immediately west of it have accommodated dextral transtension, of which the dextral component has been interpreted as resulting from the oblique convergence between the Pacific and Eurasian plates[42,43]. We note, however, that model $I_{INT}$-R also shows signs of dextral transtension near the Western Pacific subduction boundary along the Ryukyu–Japan–Kuril–Kamchatka segment.

The conclusion that the Asian tectonic evolution is best characterized by $I_I = 1740 \pm 300$ km ($R = \sim 0.40$–0.52) has important implications for domain boundary migrations and upper mantle volume fluxes in the eastern hemisphere. With $I_I = 1740$ km, the total upper mantle volume flux induced by the advancing Indian slab is roughly $3.5 \times 10^9$ km$^3$ over 52 Myr, while those of the retreating Western Pacific slabs and Sunda slab are roughly $2.0 \times 10^9$ km$^3$ and $1.7 \times 10^9$ km$^3$ over ~52 Myr, respectively (Fig. 1c). It implies that, since ~52 Ma the upper mantle domain beneath East Asia has been growing at an average rate of ~100 m$^3$ s$^{-1}$, the Indian upper mantle domain has been growing at ~1100 m$^3$ s$^{-1}$, while the Pacific upper mantle domain has been shrinking at ~1200 m$^3$ s$^{-1}$. We propose that, to accommodate the expansion in easternmost Asia, mantle material north of the Indian slab has moved eastward, consistent with East Asian geochemistry of Cenozoic magmatic rocks showing a Dupal signature[44] and shear-wave splitting observations showing an overall approximately east–west trend below East China[45,46] (Fig. 1b, c) and in agreement with the general mantle flow pattern observed in our models (Fig. 7). Locally, Pacific mantle material has likely infiltrated the Asian domain through slab windows below northernmost Kamchatka and the Taiwan–Philippines region (Fig. 1b, c).

The hole left by Indian indentation and slab advance has been partly filled by Burma–Sunda slab-rollback-induced mantle flow and by toroidal mantle return flow around the eastern and western Himalayan syntaxes, as suggested by shear-wave splitting observations[45].

We note that shear-wave splitting observations imply a tighter toroidal flow around the eastern syntaxis than observed in the models. This could be because of two reasons. First of all, it is likely that there is a small slab window just south of the eastern syntaxis, because the Arakan slab only continues to ~26.5° north[47], while the northern edge of the eastern syntaxis is at ~28° north. In addition, there is likely a slab window between the Arakan slab and the Andaman slab (between ~15° and ~20° north) and the Arakan slab is likely also segmented and torn[47]. Such slab windows/gaps/tears would allow mantle material to flow from the Tibetan side (higher dynamic pressure), around the eastern syntaxis and towards the Indian side south of the Himalayan slab (lower dynamic pressure), as illustrated by the orange arrow in Fig. 1b. The orientation of flow is consistent with shear-wave splitting results[47]. In the analogue models, there is no slab window present in the Indian indenter near the eastern syntaxis, so inflow into the region south of the Indian indenter front is not possible and so toroidal return flow is more limited. A second reason could be that the analogue models use a linear viscous (Newtonian) rheology for the sub-lithospheric mantle, while the sub-lithospheric mantle in nature is possibly dominated by a non-linear power law rheology[48]. Such a power law rheology enhances strain localization thereby promoting tighter, more localized, toroidal return flow patterns around lateral slab edges.

To return to the topic of mantle flow in the Indian domain, we propose that inflow into the Indian domain is also partly accommodated by southwest to west-directed Pacific mantle flow from the Philippine Sea domain towards New Guinea and northern Australia south of and sub-parallel to the Java–Banda slab wall (Fig. 1b, c). This is consistent with shear-wave splitting observations[49]. Our models thus imply large-scale exchange and mixing of mantle material between the Indian, Asian and Pacific domains along a clockwise circulation pattern, leaving an imprint on the structure, dynamics and chemistry of crust and mantle.

## Methods
**Experimental set-up**. We use four-dimensional analogue laboratory experiments of Asian deformation, as they allow us for the first time to integrate several geodynamic processes including continental indentation, subduction rollback, mantle flow and continental gravitational spreading on a very large spatial scale. The experiments provide us a means to quantify and reproduce the surface strain field, displacement field, mantle flow patterns and topography at an unprecedented large scale (representing thousands of kilometres) and at the same time allow us to simulate small-scale (representing kilometre scale) strain localization through faulting and shearing.

The experiments consist of a layered rheological system, with a brittle top layer and underlying high-viscosity layer representing the continental lithosphere and a low-viscosity bottom layer representing the sub-lithospheric mantle, following earlier experimental work on continental lithospheric deformation and backarc extension[26,50]. The lithosphere is confined within an internal rectangular compartment, 100 cm × 110 cm horizontally, that is located within a larger rectangular box, 130 × 150 cm, that is filled with the sub-lithospheric mantle (Fig. 2). At sub-lithospheric mantle depth, the internal compartment (representing the Eurasian domain) and the outer domain (representing the oceanic domains) are connected through a window in the southeastern corner (representing a slab window zone in Southeast Asia) such that the continental lithosphere inside the internal compartment is isostatically supported by the sub-lithospheric mantle and is in isostatic equilibrium with the oceanic domains.

Continental deformation in the Central, East and Southeast Asian lithosphere is enforced through three externally driven boundaries to simulate approximately northward Indian indentation, approximately southward rotational rollback of the Sunda subduction zone and approximately eastward translational–rotational rollback of the Western Pacific subduction zones (Fig. 2). Considering that we impose velocities, our models are not fully dynamic (exclusively buoyancy-driven) but include both kinematically imposed forces and internal buoyancy forces. The velocities of the three boundaries can be set individually by three individual step-motors. The northern (leading) boundary of the Indian intender block represents the Indian subduction hinge and slab, and so the advance velocity of this block, which is determined by step-motor 1, is exactly the advance velocity of the subduction hinge and slab. The retreating boundaries that are connected to step-motors 2 and 3 represent the retreating slabs and subduction hinges of the Sunda and Western Pacific subduction margins, respectively, and their velocities represent the retreat velocities of the subduction hinges and slabs. The boundary velocity values for the different experiments discussed in the text are listed in Table 1.

Our models do not include subduction zone plate boundary-induced simple shear, as is also the case for earlier models of India–Asia indentation and Asian deformation[9,11,12,19]. We note that the simple shear at the continental and oceanic subduction boundaries impose a local deformation on the overriding plate, as the length-scale of the deformation induced by the simple shear scales with the down-dip length of the subduction zone interface (e.g. see ref. [51]), which is of the order 100–200 km. As such, the simple shear at the subduction boundaries is not significant for the large-scale (thousands of km), widespread deformation of Central, East and Southeast Asia that is the focus of this study.

For the current study, we have chosen analogue experiments as our geodynamic modelling method, as they have a number of advantages for the purpose of this study compared to other geodynamic modelling methods, namely: (1) The analogue models are inherently four-dimensional (three-dimensional space+time). (2) The analogue models can be conducted at a massive scale (e.g. continental lithosphere of 8000 km × 8800 km), while still retaining enough resolution at small scale to simulate shear localization and brittle faulting in the upper part of the continental lithosphere. (3) The models have a free top surface such that they can produce mountain topography and morphology that are comparable to that in the Tibet–Himalaya region, as well as basin bathymetry and morphology that are comparable to the marginal basins in East and Southeast Asia.

**Scaling of experiments and experimental materials**. The experiments are scaled for gravity such that they fulfil the requirements of geometrical, kinematic and dynamic similarity with respect to the natural prototype[52–57]. The scaling parameters are listed in Table 2. Previous models simulating Asian deformation that were scaled for gravity and were isostatically supported by an asthenosphere[12,19,20] used smaller scaled lengths than the current study. In the current experiments, a length-scale factor of $1.25 \times 10^{-7}$ is applied (1 cm represents 80 km) such that the Indian indenter width (east–west) of 30 cm represents 2400 km in nature (approximate width of Indian indenter), a 50 cm retreating Sunda subduction boundary represents 4000 km, a 100 cm Western Pacific retreating subduction boundary represents 8000 km and a continental lithosphere thickness of 1.3 cm represents 104 km. These scaled model dimensions are of comparable magnitude as those in nature at the onset of collision at ~50 Ma (Fig. 1c). The sub-lithospheric mantle layer is 7.7 cm thick, representing 616 km in nature. Each experiment lasts 40 h, which represents ~52 million years in nature, giving a model/nature timescale factor of $8.781 \times 10^{-11}$. We note that, in our models, the thickness of the Asian continental lithosphere and the thickness of its individual layers are constant at the start of each model run. It is therefore evident that we did not consider any potential lateral variations in lithospheric thickness caused by earlier phases of deformation of the Asian lithosphere.

In the experiments, we scale for density contrasts[56] and we adopt a constant density for the continental lithosphere of $1232 \pm 7$ kg m$^{-3}$ and a constant density for the sub-lithospheric mantle of $\rho_{SLM} = 1426 \pm 2$ kg m$^{-3}$, giving a density contrast $\Delta\rho(\text{model}) = 194 \pm 9$ kg m$^{-3}$. This is equivalent to the density contrast in nature of $\Delta\rho(\text{nature}) = 194$ kg m$^{-3}$ assuming a sub-lithospheric mantle density of 3250 kg m$^{-3}$, a 40-km-thick crust with density 2745 kg m$^{-3}$ and 64-km-thick lithospheric mantle with density 3250 kg m$^{-3}$. Scaling for density contrasts requires that for the scaling of surface topography we need to apply a correction factor $C_{Topo}$ as discussed in ref. [56], with $C_{Topo} = \rho_{SLM}(\text{model})/\rho_{SLM}(\text{nature}) \approx 0.44$.

The rheological layering of the continental lithosphere and underlying sub-lithospheric mantle in the experiments is achieved using different materials with different rheological behaviour, following earlier work[12,19,26,50,52]. The 1.3-cm-thick-layered model lithosphere (0.3 cm brittle top and 1.0 cm viscous bottom) represents a simplified layered Asian continental lithosphere with a 24-km-thick brittle top and an 80-km-thick viscous bottom.

The brittle top layer of the continental lithosphere consists of fine-grained glass microspheres (grain size = 90–180 μm) mixed with hollow glass microspheres to attain the correct density. Granular materials such as sand and glass microspheres have a frictional-plastic (brittle) rheology that follows the Mohr–Coulomb failure criterion[58–60], show strain localization[61] and display strain weakening[58]. The microspheres used here have a coefficient of internal friction $\mu = 0.65$ at peak strength[59], which falls in the natural range (0.49–1.00)[59], are well rounded, have a high sphericity and have a very low cohesion of 0–15 Pa, which scales to values of 0–120 MPa in nature, in accordance with reported values for the natural prototype (15–135 MPa)[59].

The viscous lower layer of the continental lithosphere consists of a high-viscosity silicone oil mixed with fine-grained iron powder to attain the right density. The silicone is a polydimethylsiloxane that has been used frequently in lithospheric and mantle scale laboratory experiments[12,19,26,50,52,62–64] and has a Newtonian viscosity at experimental strain rates (<10$^{-2}$ s$^{-1}$)[65,66]. The mix has a dynamic shear viscosity of $5.8 \pm 0.2 \times 10^4$ Pa s.

**Table 1 Experimental and scaled displacements and displacement rates of the three plate boundaries**

| Experiment | Plate boundary | Advance/ retreat [cm] | Advance/retreat rate [cm h$^{-1}$] | Scaled advance/ retreat [km] | Scaled advance/retreat rate[a] [cm year$^{-1}$] | $R$ ($v_I - v_{WP}$)/ ($v_I + v_{WP}$) |
|---|---|---|---|---|---|---|
| I$_{MIN}$-R (minimum indentation, with rollback) | India | 13.12 | 0.328 | 1050 | 2.02 | 0.25 |
| | Western Pacific | 7.95 | 0.199 | 636 | 1.22 | |
| | Sunda | 11.00 | 0.275 | 880 | 1.69 | |
| I$_{INT}$-R (intermediate indentation, with rollback) | India | 23.60 | 0.590 | 1888 | 3.63 | 0.50 |
| | Western Pacific | 7.95 | 0.199 | 636 | 1.22 | |
| | Sunda | 11.00 | 0.275 | 880 | 1.69 | |
| I$_{MAX}$-R (maximum indentation, with rollback) | India | 34.10 | 0.852 | 2728 | 5.25 | 0.62 |
| | Western Pacific | 7.95 | 0.199 | 636 | 1.22 | |
| | Sunda | 11.00 | 0.275 | 880 | 1.69 | |
| I$_{INT}$-NR (intermediate indentation, no rollback) | India | 23.60 | 0.590 | 1888 | 3.63 | 1.00 |
| | Western Pacific | 0 | 0 | 0 | 0 | |
| | Sunda | 0 | 0 | 0 | 0 | |
| NI-R (no indentation, with rollback) | India | 0 | 0 | 0 | 0 | −1.00 |
| | Western Pacific | 7.95 | 0.199 | 636 | 1.22 | |
| | Sunda | 11.00 | 0.275 | 880 | 1.69 | |
| I-R$_{MO}$ (maximum indentation, with rollback) (no brittle top) | India | | 0.852 | | 5.25 | 0.62 |
| | Western Pacific | | 0.199 | | 1.22 | |
| | Sunda | | 0.275 | | 1.69 | |

[a]This is calculated from the scaled advance/retreat and the timing of onset of India–Eurasia collision, which we take as 52 Ma[3]

**Table 2 Experimental scaling parameters**

| Parameter | Notation | Dimensions | Experiment | Nature | Scaling factor (experiment/nature) |
|---|---|---|---|---|---|
| Gravity | $g$ | [m s$^{-2}$] | 9.8 | 9.8 | 1 |
| Thickness | | | | | $1.25 \times 10^{-7}$ |
| Continental lithosphere | $T_{CL}$ | [m] | 0.013 | $1.04 \times 10^5$ | |
| Sub-lithospheric mantle | $T_{SLM}$ | [m] | 0.08 | $6.40 \times 10^5$ | |
| Density | | | | | |
| Continental lithosphere | $\rho_{CL}$ | [kg m$^{-3}$] | 1232 ± 7 | 3056 | |
| Sub-lithospheric mantle | $\rho_{SLM}$ | [kg m$^{-3}$] | 1426 ± 2 | 3250 | |
| Density contrast ($\rho_{SLM} - \rho_{CL}$) | $\Delta\rho$ | [kg m$^{-3}$] | 194 | 194 | 1 |
| Time | $t$ | [s] | 3600 | $4.100 \times 10^{13}$ | $8.781 \times 10^{-11}$ |
| Coefficient of internal friction (brittle lithosphere) | $\mu$ | | 0.65 | 0.65 | 1 |
| Cohesion (brittle lithosphere) | C | [Pa] | 0–15 | 0–$1.2 \times 10^8$ | $1.25 \times 10^{-7}$ |
| Viscosity | | | | | $1.1 \times 10^{-17}$ |
| Viscous lithosphere | $\eta_{CL}$ | [Pa s] | $5.8 \pm 0.2 \times 10^4$ | $5.3 \times 10^{21}$ | |
| Sub-lithospheric mantle | $\eta_{SLM}$ | [Pa s] | 254 ± 7 | $2.31 \times 10^{19}$ | |
| Topography | $h$ | [m] | 0.001 | 3520 | $1.25 \times 10^{-7}/C_{Topo} = 2.84 \times 10^{-7}$ |

The bottom layer representing the sub-lithospheric mantle consists of glucose syrup, which is a Newtonian viscous material[67], with a dynamic shear viscosity of 254 ± 7 Pa s at 20 ± 0.5 °C, which is the temperature at which the experiments are conducted. With a viscosity scaling factor of $1.1 \times 10^{-17}$, the experimental sub-lithospheric upper mantle viscosity represents $2.3 \times 10^{19}$ Pa s in nature, which is in general accordance with values estimated previously[68–70], which are of the order $10^{19}$–$10^{20}$ Pa s.

Dynamic scaling requires that the experiments are conducted in the same flow regime as in nature[56] (laminar flow regime with dihedral symmetry across two orthogonal planes), which demands that the Reynolds number Re << 1, with Re = $\rho v d/\eta$. Here $\rho$ is the density of the ambient fluid (1426 kg m$^{-3}$ for the sub-lithospheric mantle), $v$ is the characteristic velocity (we take the velocity of the Indian indenter for experiment I$_{INT}$-R, $v_I = 1.6 \times 10^{-6}$ m s$^{-1}$), $d$ is the characteristic length scale (we take the width of the Indian indenter, $d = 0.30$ m) and $\eta$ is the dynamic shear viscosity of the ambient fluid (254 Pa s for the sub-lithospheric mantle). This gives an experimental Re = $2.7 \times 10^{-6}$ << 1.

Considering that the experiments are scaled for gravity, that the Asian continental lithosphere is isostatically supported and that there are lateral density differences between the continental and oceanic domains, there are elevation differences and gravitational potential energy differences in the experiments. The average density of the continental Asian lithosphere is lower than that of the oceanic domains, and therefore its elevation and potential energy are higher than that of the surrounding oceanic domains. With a length scaling factor of $1.25 \times 10^{-7}$ in our models and a $C_{Topo} = 0.44$, we calculate that the elevation of the undeformed continental lithosphere with respect to the oceanic domains scales to

6.2 km. This is effectively the same as the elevation difference in nature (~6 km) with an average ocean basin depth in the Western Pacific and Northeast Indian Ocean of ~5–6 km below sea level and an average continental elevation of ~0–1 km above sea level.

**Recording of experiments**. The experiments have been recorded with a particle image velocimetry (PIV) system using four digital cameras. Camera one provided a top view overview of the entire experiment in order to record the progressive evolution of the surface structures and to extract the evolution of the surface strain field and displacement field. Cameras two and three were set up in stereoscopic arrangement (sPIV) in order to extract the progressive evolution of the surface topography of part of the experimental surface area. Camera four provided an oblique view to record the progressive evolution of the surface structures. Details on PIV and sPIV recordings for flow, surface strain and surface topography can be found in earlier works[63,71–73].

**Rates of advance and retreat**. We test the role of the advance rate (indentation rate) of the India–Eurasia collisional boundary, i.e. the rate of migration of the Indian continental subduction zone hinge and slab towards the overriding Eurasian plate, on the style and extent of widespread continental deformation in Central, East and Southeast Asia. This advance rate is varied with respect to the rollback (retreat) rates of the Western Pacific and Sunda subduction zones, which are kept constant except for experiment I$_{INT}$-NR where the rollback rates are zero. We keep these Western Pacific and Sunda rollback rates constant because they are better

constrained from reconstructions, total extension estimates and seismic tomography[26,74–78] than the Indian hinge advance rate, which is less constrained due to uncertainty in the total amount of Asian shortening. With this analogue modelling approach, we can, for the first time, provide insight into the possible dynamic interaction between the Pacific and Sunda subduction zones and the collision zone and how this has shaped the Asian landscape and interior. The advance rate can be derived from the amount of shortening that has been accommodated in the overriding plate since the start of collision between India and Eurasia, which we assume to have occurred at 52 Ma and is within the range of 52–55 Ma proposed recently[3]. Total shortening estimates range from a minimum of ~1000 km to a maximum of 1800–2000 km[2,3]. These estimates provide geologically constrained lower and upper bounds for the advance rate of 2.0 and 3.6 cm year$^{-1}$, respectively. We also test Asian deformation with an absolute maximum advance rate of 5.2 cm year$^{-1}$ assuming that the India–Eurasia convergence rate (some 5–6 cm year$^{-1}$ averaged over the past 52 Myr[6]) is accommodated entirely by indentation of Eurasia. Finally, we have run several other experiments (not shown), one of which tested Asian deformation without India–Asia indentation (Indian advance rate = 0 cm year$^{-1}$). This experiment is dominated by rift and graben structures, most of them striking (sub)parallel to the Pacific subduction boundary in the east and northeast and sub-parallel to the Sunda subduction boundary in the south, as well as a significant number of extensional structures that have an oblique or perpendicular strike with respect to the subduction boundaries.

**East–west extension in Asia.** The approximate range of values for the total amount of WNW–ESE extension in northern East Asia along profile 1 in Fig. 1b due to Cenozoic extensional faulting and backarc basin opening is presented in Fig. 6a, c. This range is based on 31–56 km of extension in the Japan arc[38] (we choose 40 km ± 15 km), 500 km (±100 km) of Japan Sea opening based on tectonic reconstructions for the region[76,77], an estimated 10 km (±5 km) for the narrow Yilan Yitong graben and 30 km (±10 km) of extension in the Baikal rift zone[79]. This gives a total extension of 580 km (±130 km).

The approximate range of values for the total amount of WNW–ESE extension in central East Asia along profile 2 in Fig. 1b due to Cenozoic extensional faulting and backarc basin opening is presented in Fig. 6a, c. This range is based on estimates of relatively minor extension of 5 km (±5 km) for each of the following rifts/basins: Hetao–Yinchuan rift, Shanxi rift, Southwest Bohai basin, and the region between the Southwest Bohai Basin and East China Sea (giving a sub-total of 20 km ± 20 km); 40 km (±10 km) for the East China Sea margin; and 60 km (±15 km) for the Okinawa Trough. These last two estimates are based on the present-day average crustal thickness in the East China Sea margin (27 km) and Okinawa Trough (18 km) and the average crustal thickness of the undeformed margins (30 km) as derived from a regional crustal thickness map[80] and the extension implied by this crustal thickness difference. This gives a total extension of 120 km (±45 km).

The approximate range of values for the total amount of east–west extension in Tibet due to Cenozoic normal faulting along approximately north–south to NNE–SSW trending grabens and rifts is presented in Fig. 6d. This range is based on earlier published estimates, which are of the order 20 km[28], ≤40 km[2] and 50–70 km[33].

## Data availability

All the data generated by the laboratory experiments that are necessary to evaluate this work are included in this published article. All the data and information that are required to reproduce the laboratory experimental results are presented in this published article.

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

## Acknowledgements

We thank G. Lister for helpful discussions and comments. W.P.S. would also like to thank Casper Schellart for help with producing Fig. 4. W.P.S. has been funded by a Vici Fellowship (016.VICI.170.110) from the Dutch National Science Foundation (NWO) and a Future Fellowship (FT110100560) from the Australian Research Council. J.C.D. and F.M.R. acknowledge funding from FCT project UID/GEO/50019/2019-IDL. J.C.D. acknowledges funding from exploratory project grant ref. IF/00702/2015.

## Author contributions

W.P.S., V.S. and J.C.D. designed the experimental apparatus, while W.P.S., Z.C. and V.S. conducted the experiments. W.P.S. conceived the ideas and wrote the first draft of the manuscript. All the authors contributed towards analysing and interpreting the experiments and towards editing of the paper.

## Competing interests

The authors declare no competing interests.
