## [Peer Review File · Nature Communications]

Reviewers' comments:

Reviewer #1 (Remarks to the Author):

This is a fine contribution from Schellart et al. I recommend publication after some very minor revisions. I note that I am not an expert on the kinds of modeling the Authors have performed herein, so I confine my comments to aspects of the tectonic problem that I do understand.

1. The Authors start the modeling at a nominal 52 Ma, but I note that the opening of the South China Sea basin occurred a bit later, perhaps really getting underway around 32-30 Ma. The models seem to indicate that rifting in the SE portion (Sunda-Sumatra) starts early and attains a maximum early on - is that correct, or I am reading the Figures incorrectly? If so, can the Authors explain the potential discrepancy in timing of South China Sea opening?

2. The upper mantle flow fields shown in Figure 7 don't reproduce the scale of the observed (shear wave splitting) flow field, which is much tighter around the Eastern Syntaxis. Any ideas why that might be? If so, can you include a brief discussion in your revised work?

3. Very nice to see the mass balance calculation for the upper mantle flow field!

4. I wonder if the Authors are aware of my work on the seismic anisotropy/upper mantle flow around the Arakan slab (Russo, Geosphere, 2013)? I'd be interested to see a detailed comparison of the modeled flow field in this region versus the actual and the possible explanations for current E-W observed shear wave splitting fast polarization directions in Sundaland -- obviously NOT for inclusion in this work, but as perhaps food for thought for the Authors should they continue modeling this type of mantle extrusion and tectonic escape in greater detail.

5. Very minor: Chen et al. appears in the reference list twice, as Ref #22 and #52. I did not check the references carefully, but that did catch my eye.

R. M. Russo
Gainesville, FL
25 April 2019

Reviewer #2 (Remarks to the Author):

Review Report on "Pacific subduction control on Tibetan extension, eastward extrusion tectonics and continental deformation in East Asia – by Schellart et al."

The paper presents a geodynamic model to show that the subduction of Pacific and Sunda plates have a significant role in deforming the entire East Asia, leading to thrusting, normal faulting, strike-slip faulting and associated eastward extrusion. From their model results, authors conclude that a critical amount of Indian indentation is essential for explaining the opening of backarc basins in East Asia (e.g., Japan Sea, Kuril Basin) along with observed extension in the central East Asia.

The continental deformation in the East Asia is indeed a lively topic of geodynamics in the framework of India-Asia collision. I am happy to see that authors have carried out very robust laboratory experiments to address this issue. This manuscript presents a comprehensive three-dimensional model of synchronous subduction of Sunda and Pacific plates and India-Asia collision system to understand the formation of diversely oriented extensional features and strike-slip faults in the East Asia continent. Finitely deformed models display significant consistency with the present-day configuration of structural elements in the East Asia. This shows the strength of the modelling approach. I consider these results are very significant contributions for understanding the kinematics of overall India-Asia collisional system. The most striking part of this modelling is to

bring out a unifying system for explaining the eastward extrusion and major fault system in the central Asia. However, there always remains some constraints in the modelling analysis. For example, this model result did not consider the deformation in the overriding plate prior to the collisional event. This paper will definitely attract readers from all groups of geosciences working in the field of collisional tectonics. Based on model results and its expected impacts within geosciences community from Structural Geology & Tectonics to Mantle petrology, I strongly recommend the paper for publication in Nature Communication.

In the context of modelling approach, authors were very careful regarding the scaling issues. Although the use of kinematic boundary conditions at Indian plate hinge as well as in the Sunda and Western Pacific Trench might develop some artifacts in the model results, consideration of R (indentation rate of Indian plate / retreat rate of Sunda and Pacific plate) for estimating the nature of continental deformation in the East Asia is a good approach. However, the model cannot be considered as self-consistent. Author's prediction of Indian indentation (1740 ± 300 km) is very close to experiments with intermediate indentation. But according to experimental results, strike slip faults along the eastern boundary of the Indian plate are more prominent with maximum indentation. In natural situations, strike slip faults are significantly active at present along the Indo-Burma range. Does it signify to have variable rate of indentation over last 52 Ma? Another point, I think author should clarify regarding the relationship between the velocity of step-motor 1 and advance rate of subduction hinge in experimental results. I am not sure if they are same.

Regarding the text, it would be nice to see a paragraph showing major outcomes of the work for explaining the deformation East Asia. Finally, I request to rephrase the sentence "The models necessitate 1740 ± 300 km ----- ~ 75 - 165 km, and extrusion is limited to ~ 260 - 360 km." in the abstract.

Reviewer #3 (Remarks to the Author):

Comments on the paper by Schellart et al.: Pacific subduction control on Tibetan extension, eastward extrusion tectonics and continental deformation in East Asia.

Long time ago, it has been shown that a large part of the deformation suffered by the Eurasian plate lithosphere was caused by India-Asia continental collision. Not only the deformation responsible for the high reliefs of the Himalayas, but also the far field deformation regimes observed very far from the indenter, the Tibetan Plateau, major strike-slip faults, extensional basins and intracontinental mountain belts. The impact of this discovery was enormous as it suggested that widespread deformation can affect wide areas in the plates interior and as a consequence showed that plates cannot be considered as rigid and undeformable. More recently, the role of slab rollback on the dynamics of subduction zones has been emphasized in many areas on earth.

The original idea highlighted in this paper combines both concepts to account for the complex structures and dynamic evolution of that emblematic deformation area, in Asia. Using scaled geodynamic experiments realized at the Kuenen-Escher Geodynamics Laboratory (Vrije University), authors demonstrate that the Pacific and Sunda subduction zones played an active role on the deformation of the Eurasian lithosphere during India-Asia convergence and continental collision. They show that only the synchronous activity and interaction of the collision zone and subduction zones can explain the entire East Asian deformation field, and demonstrate that enigmatic east-west extension in Tibet, eastward continental extrusion and backarc basin formation along the East Asian margin are controlled by large-scale Pacific slab rollback. In addition, experiments give a dynamic view of deep mantle circulation below that part of Eurasia. This is a significant step forward in our knowledge of plate tectonics.

Constraints from the models allows a quantitative estimate of the relative role of India indentation relative to slab rollback driven Asian extrusion. The manuscript is concisely written and well

illustrated. The complex experiments are well conducted and scaling is realistic. experimental results are significant and the analysis well done. Referencing is fair, consistent and supported by useful supplementary information..

This paper represents a major contribution for the knowledge of our planet dynamics. It must be published as it stands in Nature Geoscience.

Jacques Malavieille

Reviewer #4 (Remarks to the Author):

•What are the major claims of the paper?

The paper claims to model the tectonics of the India-Asia collision and the subduction tectonics of the east and southeast Asian margins, via analog experiments. It is claimed that subduction zone rollback plays a hitherto recognised role in controlling East Asian continental tectonics, including the late Cenozoic ~E-W extension of the Tibetan Plateau.

•Are the claims novel? If not, please identify the major papers that compromise novelty

The analog model is novel, to my knowledge. The claims are not. For example, the abstract states that "The prevailing explanation considers the role of the Pacific and Sunda subduction zones as passive during Asian deformation". Other papers have previously considered the role of rollback of these subduction zones. For example:

Watson et al 1987

Royden et al 2008

Capitanio et al 2015

- etc etc. A quick online search shows up >90 papers with the title or abstract containing "Pacific" "Asia" and "rollback".

Schellart has worked on similar themes before, and should not be left out of the list, e.g. Schellart et al 2003, 2005 (cited) and others, including Schellart et al 2006 (not cited).

•Will the paper be of interest to others in the field?

Yes, it is a major field

•Will the paper influence thinking in the field?

Not so likely. As noted above, the overall claim is not novel.

•Are the claims convincing? If not, what further evidence is needed?

A major issue of the paper is that the model does not involve subduction, so that the role of the subducted Indian Plate is not considered (e.g. Capitanio et al., 2010; Jolivet et al 2018). This issue also applies to the Pacific and Sunda subduction zones. Arguably the fate of the slab is less important in the regional dynamics if an oceanic subduction zone undergoes rollback and back-arc extension, as was the case for the Early Cenozoic. But in the late Cenozoic the Pacific margin experienced periods of compressional tectonics, partly caused by the convergence of Australia and Asia (Hall, 2002, 2011). This deformation is not included in the Schellart et al paper, which seriously affects its claim to represent geological reality. For example, the Neogene -E-W extension of the Tibetan Plateau takes place during this compression (see Royden et al 2008), which throws the whole validity of the Schellart et al models into doubt. (NB It is fairly clear that Tibetan rifting is a relatively young phenomenon, from 15-0 Ma. There are claims for earlier extension based on the N-S orientation of dykes within Tibet, cited in this ms, but these data indicate the relative stress field, and do not show that there was a horizontal extensional stress large enough to cause brittle faulting in the crust).

The models in the paper predict major left-lateral shear through central and NE Asia. This does not correspond to reality, where the shear sense in east Asia was predominantly right-lateral through the Cenozoic (e.g. Jolivet et al., 1990; Ren et al 2002). This subduction obliquity has been proposed to have a major role in the construction of the marginal basins (e.g. Yin 2010). The models don't consider other factors that could have been major influences on Tibetan tectonics, like lower crustal flow, and loss of the lower lithosphere.

•Are there other experiments that would strengthen the paper further? How much would they improve it, and how difficult are they likely to be?

These experiments don't involve subduction, either continental or oceanic, so they are at a disadvantage compared with numerical models.

•Are the claims appropriately discussed in the context of previous literature?

No, see above. There are too many points where the model does not match the geology.

•If the manuscript is unacceptable in its present form, does the study seem sufficiently promising that the authors should be encouraged to consider a resubmission in the future?

No, unfortunately. The models give a partial simulation of the upper crustal fault patterns across Asia, but it is too ambitious to claim that they shed light on the overall tectonics and dynamics.

Response to reviewers

Reviewers' comments:

Reviewer 1:

Reviewer #1 (Remarks to the Author):

This is a fine contribution from Schellart et al. I recommend publication after some very minor revisions. I note that I am not an expert on the kinds of modeling the Authors have performed herein, so I confine my comments to aspects of the tectonic problem that I do understand.

1. The Authors start the modeling at a nominal 52 Ma, but I note that the opening of the South China Sea basin occurred a bit later, perhaps really getting underway around 32-30 Ma. The models seem to indicate that rifting in the SE portion (Sunda-Sumatra) starts early and attains a maximum early on - is that correct, or I am reading the Figures incorrectly? If so, can the Authors explain the potential discrepancy in timing of South China Sea opening?

It is true that spreading in the South China Sea started at ~33 Ma [Li et al., 2014], but the extensional phase of the South China Sea region that preceded the spreading phase started earlier. In particular, at the southern passive margin of the South China Sea extension started at ~55 Ma [Wang et al., 2015] and along the northern margin in the Quiondongnan Basin, for example, at ~56 Ma [Wu et al., 2016] (Fig. 1a). these times correspond well with the start of the models at ~52 Ma. It is also true that the rifting in the SE portion (Sundaland region) starts early in our models that include slab rollback, but the maximum is reached towards the end of the models, as can be seen for experiment I_{INT}-R in Supplementary Fig. 5 and for experiment I_{MIN}-R in the new Supplementary Fig. 6, which is better in agreement with observations. To accommodate the comment from the reviewer and to prevent potential confusion, we have added a paragraph of text at the end of the section "Experimental results of Asian deformation"

to explain the above and have added an additional figure to the Supplementary Information (Fig. S6) showing four evolutionary stages of experiment I_{MIN}-R.

2. The upper mantle flow fields shown in Figure 7 don't reproduce the scale of the observed (shear wave splitting) flow field, which is much tighter around the Eastern Syntaxis. Any ideas why that might be? If so, can you include a brief discussion in your revised work?

Yes, good point. Shear wave splitting observations show a tighter toroidal flow around the eastern syntaxis. This could be because of two reasons: (1) It is likely that there is a small slab window just south of the eastern syntaxis, because the Arakan slab (also referred to as Burma slab, see point 4 by reviewer 1) only continues to ~26.5° north [Russo, Geosphere 2012], while the northern edge of the eastern syntaxis is at ~28° north. In addition, there is likely a slab window between the Arakan slab and the Andaman slab (between 15 and 20° north) and the Arakan slab is likely also segmented and torn [Russo, Geosphere 2012]. Such slab windows/gaps tears would allow mantle material to flow from the Tibetan side (higher dynamic pressure), around the eastern syntaxis and towards the Indian side south of the Himalayan slab (lower dynamic pressure), as illustrated by the orange arrow in Fig. 1b. The orientation of flow is consistent with the shear-wave splitting results presented in Russo [2012]. In the analogue models there is no slab window present in the Indian indenter near the eastern syntaxis, so inflow into the region south of the Indian indenter front is not possible and so toroidal return flow is more limited. (2) the analogue models use a linear viscous (Newtonian) rheology for the sub-lithospheric mantle, while the sub-lithospheric mantle in nature is likely dominated by a non-linear power-law rheology [Mackwell et al., GRL 1990]. Such a power law rheology enhances strain localization and generally tighter, more localized, toroidal return flow patterns around lateral slab edges [Jadamec and Billen, Nature 2010]. To accommodate the comment from the reviewer, we have included part of the text from above in the revised manuscript in the section "Implications for mantle flow".

3. Very nice to see the mass balance calculation for the upper mantle flow field!
Thanks very much for the positive words!

4. I wonder if the Authors are aware of my work on the seismic anisotropy/upper mantle flow around the Arakan slab (Russo, Geosphere, 2013)? I'd be interested to see a detailed comparison of the modeled flow field in this region versus the actual and the possible explanations for current E-W observed shear wave splitting fast polarization directions in Sundaland -- obviously NOT for inclusion in this work, but as perhaps food for thought for the Authors should they continue modeling this type of mantle extrusion and tectonic escape in greater detail.

We were not aware of this interesting work (note that we think reviewer 1 refers to [Russo, Geosphere 2012], because we could not find [Russo, Geosphere 2013]). As reviewer 1 notes, a detailed comparison between our work and that of Russo [Geosphere 2012] is not for inclusion in this work. However, we did find this work very useful for our discussion and our response to comment 2 from reviewer 1 (see above). Indeed, it provides a nice explanation for the tighter return flow around the eastern syntaxis in nature as implied by shear wave splitting (Fig. 1c) than what we observe in our models (e.g. Fig. 7).

5. Very minor: Chen et al. appears in the reference list twice, as Ref #22 and #52. I did not check the references carefully, but that did catch my eye.

Thanks for spotting this error. We made a mistake here. This should have been another reference [Chen et al., 2017]. We have now corrected this error and we have also checked the reference list very carefully again to check that there were no other duplicate references or missing references.

R. M. Russo
Gainesville, FL
25 April 2019

Reviewer 2:

Reviewer #2 (Remarks to the Author):

Review Report on “Pacific subduction control on Tibetan extension, eastward extrusion tectonics and continental deformation in East Asia – by Schellart et al.”

The paper presents a geodynamic model to show that the subduction of Pacific and Sunda plates have a significant role in deforming the entire East Asia, leading to thrusting, normal faulting, strike-slip faulting and associated eastward extrusion. From their model results, authors conclude that a critical amount of Indian indentation is essential for explaining the opening of backarc basins in East Asia (e.g., Japan Sea, Kuril Basin) along with observed extension in the central East Asia.

The continental deformation in the East Asia is indeed a lively topic of geodynamics in the framework of India-Asia collision. I am happy to see that authors have carried out very robust laboratory experiments to address this issue. This manuscript presents a comprehensive three-dimensional model of synchronous subduction of Sunda and Pacific plates and India-Asia collision system to understand the formation of diversely oriented extensional features and strike-slip faults in the East Asia continent. Finitely deformed models display significant consistency with the present-day configuration of structural elements in the East Asia. This shows the strength of the modelling approach. I consider these results are very significant contributions for understanding the kinematics of overall India-Asia collisional system. The most striking part of this modelling is to bring out a unifying system for explaining the eastward extrusion and major fault system in the central Asia.
Thank you very much for these positive words.

However, there always remains some constraints in the modelling analysis. For example, this model result did not consider the deformation in the overriding plate prior to the collisional event.
Indeed, it is true that we did not consider overriding plate deformation prior to collision, and the potential variation in lithospheric thickness this could have caused.
We now explicitly state this in the manuscript (Methods section).
Inherited weak structures could control the localisation of deformation. However, our models still reproduce at large the deformation pattern as observed in nature so we do not expect that structural inheritance would have a first-order impact.

This paper will definitely attract readers from all groups of geosciences working in the field of collisional tectonics. Based on model results and its expected impacts within geosciences community from Structural Geology & Tectonics to Mantle petrology, I strongly recommend the paper for publication in Nature Communication.

Thank you very much for these positive words.

In the context of modelling approach, authors were very careful regarding the scaling issues. Although the use of kinematic boundary conditions at Indian plate hinge as well as in the Sunda and Western Pacific Trench might develop some artifacts in the model results, consideration of R (indentation rate of Indian plate / retreat rate of Sunda and Pacific plate) for estimating the nature of continental deformation in the East Asia is a good approach. However, the model cannot be considered as self-consistent.

It is true that the models are not fully dynamic (self-consistent) because they use kinematic boundary conditions, as the reviewer notes. Self-consistent models would require that motions and deformation are driven entirely by buoyancy forces.

To accommodate the comment from the reviewer, we now specifically state in the Methods that our models are not fully dynamic (exclusively buoyancy-driven) but include both kinematically imposed forces and internal buoyancy forces.

Author's prediction of Indian indentation (1740 ± 300 km) is very close to experiments with intermediate indentation. But according to experimental results, strike slip faults along the eastern boundary of the Indian plate are more prominent with maximum indentation. In natural situations, strike slip faults are significantly active at present along the Indo-Burma range. Does it signify to have variable rate of indentation over last 52 Ma?

It is true that strike-slip faulting along the eastern boundary of the Indian plate is most significant in the experiment with maximum indentation (I_{MAX-R}), but I_{MAX-R} also shows very significant strike-slip faulting along the western boundary of the Indian plate. Indeed, I_{MAX-R} shows an approximately symmetrical pattern of strike-slip faulting east and west of India (Fig. 4c), which is not in agreement with observations, which show significant strike-slip faulting along the eastern boundary but limited strike-slip faulting along the western boundary. In fact, this asymmetry is best reproduced in model I_{INT-R} , which shows significant dextral, ~north-south oriented strike-slip faulting along the eastern boundary of India, and limited strike-slip faulting along the western boundary. As such, it appears to that variable rates of indentation are not required (although they might still have occurred in nature) to reproduce the first-order strike-slip faulting east and west of the India indenter.

We have included part of the text from above into the revised manuscript to accommodate the comment from Reviewer 2.

Another point, I think author should clarify regarding the relationship between the velocity of step-motor 1 and advance rate of subduction hinge in experimental results. I am not sure if they are same.

The Indian indenter block in our models is connected to step-motor 1, which determines its velocity. The northern (leading) edge of the Indian indenter block represents the Indian subduction hinge and slab, and so the advance velocity of this block, which is determined by step-motor 1, is exactly the advance velocity of the subduction hinge.

We have now clarified this in the Methods section, where we describe the model apparatus.

Regarding the text, it would be nice to see a paragraph showing major outcomes of the work for explaining the deformation East Asia.

To accommodate the comment from reviewer 2, we have added a new paragraph to the main text to summarize the results of our best fitting models and how they agree with the first order Cenozoic geological, structural, and tectonic observations in East and Southeast Asia.

Finally, I request to rephrase the sentence "The models necessitate 1740 ± 300 km -----~75-165 km, and extrusion is limited to ~260-360 km." in the abstract.

We have rephrased the sentence to improve its readability.

Reviewer 3:

Reviewer #3 (Remarks to the Author):

Comments on the paper by Schellart et al.: Pacific subduction control on Tibetan extension, eastward extrusion tectonics and continental deformation in East Asia.

Long time ago, it has been shown that a large part of the deformation suffered by the Eurasian plate lithosphere was caused by India-Asia continental collision. Not only the deformation responsible for the high reliefs of the Himalayas, but also the far field deformation regimes observed very far from the indenter, the Tibetan Plateau, major strike-slip faults, extensional basins and intracontinental mountain belts. The impact of this discovery was enormous as it suggested that widespread deformation can affect wide areas in the plates interior and as a consequence showed that plates cannot be considered as rigid and undeformable. More recently, the role of slab rollback on the dynamics of subduction zones has been emphasized in many areas on earth.

The original idea highlighted in this paper combines both concepts to account for the complex structures and dynamic evolution of that emblematic deformation area, in Asia. Using scaled geodynamic experiments realized at the Kuenen-Escher Geodynamics Laboratory (Vrije University), authors demonstrate that the Pacific and Sunda subduction zones played an active role on the deformation of the Eurasian lithosphere during India-Asia convergence and continental collision. They show that only the synchronous activity and interaction of the collision zone and subduction zones can explain the entire East Asian deformation field, and demonstrate that enigmatic east-west extension in Tibet, eastward continental extrusion and backarc basin formation along the East Asian margin are controlled by large-scale Pacific slab rollback. In addition, experiments give a dynamic view of deep mantle circulation below that part of Eurasia. This is a significant step forward in our knowledge of plate tectonics.

Thank you very much for these positive words.

Constraints from the models allows a quantitative estimate of the relative role of India indentation relative to slab rollback driven Asian extrusion. The manuscript is concisely written and well illustrated. The complex experiments are well conducted and scaling is realistic. experimental results are significant and the analysis well done. Referencing is fair, consistent and supported by useful supplementary information..

Thank you again for these positive words.

This paper represents a major contribution for the knowledge of our planet dynamics. It must be published as it stands in Nature Geoscience.

Thank you very much again for these positive words.

Jacques Malavieille

Reviewer 4:

Reviewer #4 (Remarks to the Author):

•What are the major claims of the paper?

The paper claims to model the tectonics of the India-Asia collision and the subduction tectonics of the east and southeast Asian margins, via analog experiments. It is claimed that subduction zone rollback plays a hitherto recognised role in controlling East Asian continental tectonics, including the late Cenozoic ~E-W extension of the Tibetan Plateau.

•Are the claims novel? If not, please identify the major papers that compromise novelty

The analog model is novel, to my knowledge.

We are happy to see that reviewer 4 agrees that the analogue models are novel.

The claims are not. For example, the abstract states that “The prevailing explanation considers the role of the Pacific and Sunda subduction zones as passive during Asian deformation”. Other papers have previously considered the role of rollback of these subduction zones.

Indeed, other papers have previously considered the role of rollback, but they do not represent the prevailing explanation. The papers from Molnar and Tapponnier [1975], Tapponnier et al. [1982] and Tapponnier et al. [2001] have gathered some 4700, 2700 and 2600 citations so far (e.g. Google Scholar), respectively, which to us indicates that they represent “the prevailing explanation”. The 3 papers that Reviewer 4 lists below have acquired much less citations.

For example:

Watson et al 1987

Royden et al 2008

Capitanio et al 2015

- etc etc.

A quick online search shows up >90 papers with the title or abstract containing "Pacific" "Asia" and "rollback".

It is true that there are papers on rollback of Pacific lithosphere and deformation in Asia, but those papers are generally concerned with, and focus on, much smaller scale subduction segments that roll back and affect deformation in a small-scale backarc basin. So these works focus on individual backarc basins such as the Banda Sea, Japan Sea, Okinawa Trough, Andaman Sea, etc. These studies generally do not concern the entire East Asian and Southeast Asian region as a whole, being affected by the entire Pacific subduction margin, the Sunda subduction margin and the Indian indenter.

An exception is Royden et al. [2008], but this paper is cited in the manuscript. Other exceptions are Fournier et al., [2004] and Schellart and Lister [2005], but these papers are also cited in the manuscript. Royden et al. [2008] proposed a conceptual model, but did not present geodynamic models to verify the physical plausibility of their conceptual model. Fournier et al., [2004] and Schellart and Lister [2005] presented geodynamic models, but did not include actively retreating Pacific and Sunda subduction boundaries (as we explain in our manuscript), which is one of the main novelties of our geodynamic models.

Watson et al. [1987] presented a conceptual model, not geodynamic models, and, furthermore, their concepts were focused on sedimentary basins in China. They did not investigate Southeast Asia, nor Mongolia, Japan, Korea and eastern Siberia. Thus, the focus of our study involves a much larger spatial domain and involves geodynamic models. Nevertheless, they did argue that eastward retreat of the subduction system along eastern China resulted in the development of extensional basins in East China.

Capitanio et al. [2015] presented an interesting subduction modelling study, but their focus was on a much smaller area and excluded the entire Pacific subduction margin. Indeed, their spatial domain of the Asian lithosphere only represented ~2000 km by 3000 km, their "Indian indenter" was only 1000 km wide, and their "Sunda subduction margin" only ~1700 km wide. This is much smaller than the actual dimensions of the Asian lithosphere that is deforming and that is affected by the Indian indenter, Sunda subduction zone and Pacific subduction margin, which is about 7000 km in the south and ~8000 km in the east and involves an Indian indenter that is ~2400 km wide. This very large spatial domain is what we model in our novel geodynamic experiments.

In any case, to accommodate the comment from the reviewer, we have included the works of Watson et al. [1987] and Capitanio et al. [2015] in the introduction.

Schellart has worked on similar themes before, and should not be left out of the list, e.g. Schellart et al 2003, 2005 (cited) and others, including Schellart et al 2006 (not cited).

Schellart et al. [2003] and Schellart and Lister [2005] are included in the manuscript and in the list of references, because they are relevant to this manuscript. The paper Schellart et al. [Earth-Science Reviews 2006] presents a tectonic reconstruction of the Southwest Pacific region between eastern Australia, New Zealand and the Fiji Islands, so this paper is not relevant here and is therefore not cited.

•Will the paper be of interest to others in the field?

Yes, it is a major field

•Will the paper influence thinking in the field?

Not so likely. As noted above, the overall claim is not novel.

We disagree. We present for the first time geodynamic models that include active Pacific margin slab rollback, Sunda slab rollback and India indentation and demonstrate their role in Asian continental deformation. This has never been done before. The papers cited by reviewer 4 have not done this: Watson et al. [1987] and Royden et al. [2008] presented conceptual ideas (not geodynamic models), while Capitanio et al. [2015] focused on a much smaller spatial domain and did not consider the active role of the western Pacific margin (see our response above). Other papers that we cite in the manuscript [e.g. Tapponnier et al., 1982; Davy and Cobbold, 1988; Fournier et al., 2004; Schellart and Lister, 2005] have not done this either, as explained in the manuscript.

•Are the claims convincing? If not, what further evidence is needed?

A major issue of the paper is that the model does not involve subduction, so that the role of the subducted Indian Plate is not considered (e.g. Capitanio et al., 2010; Jolivet et al 2018). This issue also applies to the Pacific and Sunda subduction zones.

We specifically state that we model the subduction hinge advance and slab advance of the Indian lithosphere (i.e. the indentation) and the subduction hinge retreat and slab rollback of the Sunda and Pacific subduction margins. So we do model the subduction zones, namely their lateral migration, but we do not model the downdip motion at the subduction zones, as also explained in significant detail in the methods section. Our models are a major step forward compared to the models of Tapponnier et al. [1982], Davy and Cobbold [1988] and Fournier et al. [2004], who only modelled the indentation of India. Capitanio et al. [2010] modelled Indian plate subduction, but did not consider the Sunda and Pacific subduction zones and, additionally, their models were limited to 2D space (2D cross-sections striking ~north-south). Furthermore, their models did not include an overriding plate, so they could not investigate Asian deformation. The recent paper from Jolivet et al. [2018] does not present geodynamic models.

Arguably the fate of the slab is less important in the regional dynamics if an oceanic subduction zone undergoes rollback and back-arc extension, as was the case for the Early Cenozoic. But in the late Cenozoic the Pacific margin experienced periods of compressional tectonics, partly caused by the convergence of Australia and Asia (Hall, 2002, 2011). This deformation is not included in the Schellart et al paper, which seriously affects its claim to represent geological reality.

We would first like to emphasize that earlier numerical modelling papers [e.g. Capitanio et al., 2015] and analogue modelling papers [e.g. Tapponnier et al., 1982; Fournier et al., 2004; Schellart and Lister, 2005] did not include such “periods of compressional tectonics”, which, according to Reviewer 4, would mean that this would also seriously affect these papers’ claims “to represent geological reality”. In any case, we do not agree with reviewer 4 on this point, as explained below.

The convergence between Australia and Asia is accommodated by subduction at the Sunda-Banda convergent margin. Only a local region in the far southeastern part of Southeast Asia (Timor-Banda region) has experienced compressional tectonics due to this convergence, and this has only occurred since ~3.5 Ma due to Australian continental passive margin-arc collision [Audley-Charles, Tectonophysics 2004], while backarc spreading and extensional tectonics in the Banda Sea took place until as recently as 3 Ma [Spakman and Hall, 2010] (Fig. 1a). Along the Pacific margin, Taiwan and northern Honshu Island in Japan have also experienced shortening, but again, these are relatively local phenomena compared to the scale of East and Southeast Asia and they have only been active since 3-2 Ma for Taiwan [Malavielle et al., GSA-SP 2002] and ~3.5 Ma for northern Honshu [Okada and Ikeda, JGR 2012]. These phases of compression and shortening are thus local and very recent, while the overall, large-scale, tectonics of East Asia during most of the Cenozoic has been dominated by extension, as shown in Fig. 1a.

To accommodate the comment from the reviewer, we have incorporated part of our response from above into the revised manuscript to emphasize the local scale and very recent occurrence of these compressive tectonic phases along the Sunda and Pacific margins, which we do not model in our experiments.

For example, the Neogene –E-W extension of the Tibetan Plateau takes place during this compression (see Royden et al 2008), which throws the whole validity of the Schellart et al models into doubt. (NB It is fairly clear that Tibetan rifting is a relatively young phenomenon, from 15-0 Ma.

The oldest extension along ~N-S striking rifts and normal faults in Tibet that has been dated so far (that we are aware of) is 19 Ma, as presented in the review paper of Li et al. [Earth-Science Reviews 2015] and the paper of Mitsuishi et al. [EPSL 2012]. Thus, this extension started much earlier than the local shortening and compression in Timor, Honshu and Taiwan, as we discuss above, which started only at 2.0-3.5 Ma. The timing of extension in Tibet

overlaps significantly with many other regions of extension in East and Southeast Asia, including the Banda Sea, Andaman Sea, Malay Basin, Nam Con Son Basin, Okinawa Trough, Shanxi graben, Weihe graben, South Ningxia region, Hetao-Yinchuan grabens, Japan Sea, Priokhotsky Rift, Baikal Rift, Kuril Basin and Kamchatka grabens. This is clearly illustrated in Fig. 1a.

There are claims for earlier extension based on the N-S orientation of dykes within Tibet, cited in this ms, but these data indicate the relative stress field, and do not show that there was a horizontal extensional stress large enough to cause brittle faulting in the crust).

The N-S oriented dikes do not indicate a relative stress field, they indicate a strain field. See for example, Williams et al. [Geology 2001] and Wang et al. [EPSL 2010]. From such a strain field one can infer a (relative) stress field. The strain field that is indicated by the dikes is east-west extension [Williams et al., Geology 2001; Wang et al., EPSL 2010], which is consistent with our modelling results. If these dikes were coincident with brittle faulting in the crust or not does not invalidate the consistency between our models and nature.

The models in the paper predict major left-lateral shear through central and NE Asia. This does not correspond to reality, where the shear sense in east Asia was predominantly right-lateral through the Cenozoic (e.g. Jolivet et al., 1990; Ren et al 2002). This subduction obliquity has been proposed to have a major role in the construction of the marginal basins (e.g. Yin 2010).

The model predictions **do** correspond to reality. Examples of major sinistral (left-lateral) shear zones in central and NE Asia are the Altyn Tagh fault, Bolnai fault and the Stanovoy sinistral shear zone. These were already shown in Fig. 1, but we have now also added names to these shear zones in Fig. 1a. These sinistral shear zones correspond very well (in location, strike orientation and shear sense) to the shear zones in our preferred model I_{INT-R} .

As for right-lateral (dextral) shear zones, the biggest dextral shear zone in East Asia is the Sakhalin-Hokkaido dextral shear zone, which formed due to asymmetric rollback of the Pacific slab, as we have already explained and modelled in Schellart et al. [Tectonics 2003], so we do not discuss that in the current manuscript. Dextral strike-slip shear zones and faults in SE Asia (Fig. 1a) are also reproduced in our models, most notably in models I_{INT-R} and I_{MAX-R} (see our response to the third comment from reviewer 2). This further shows the very good agreement between our best-fitting model and nature.

As for subduction obliquity, it has never been shown (with geodynamic models) to cause marginal/backarc basin opening. Only conceptual models have been proposed, but it has never been supported with geodynamic models. Besides, plate kinematic calculations show that Pacific-East Asia convergence was close to orthogonal from the Eocene to the present [e.g. Northrup et al., 1995; Muller et al., AREPS 2016].

In any case, we have incorporated part of the text from above into the main text of the revised manuscript to accommodate the comment from the reviewer, we have labelled several of the sinistral shear zones in central and East Asia in Fig. 1a, and we have also indicated the shear sense for several of the strike-slip faults for the models in Fig. 4.

The models don't consider other factors that could have been major influences on Tibetan tectonics, like lower crustal flow, and loss of the lower lithosphere.

It is true that our models do not consider lower crustal flow in Tibet, nor loss of lower lithosphere below Tibet. Geodynamic models that do consider such things generally focus on a much smaller scale (Himalaya-Tibet), and only simulate Himalaya-Tibetan tectonics [e.g. Royden et al., 1997; Beaumont et al., 2001; England and Houseman, 1989]. Furthermore, such geodynamic models do not consider brittle faulting, as these models only use viscous rheologies. Geodynamic models of lower crustal flow generally also lack a sub-lithospheric mantle. Our models simulate a much larger spatial domain that is about 1 order of magnitude larger, include a sub-lithospheric mantle and include brittle faulting.

In any case, to respond to the comment from the reviewer, we have added text to the section "The role of Pacific subduction in driving east-west extension in Tibet", explaining that for east-

west extension in Tibet lower crustal flow or loss of the lower lithosphere are not essential processes, as demonstrated by our experiments.

•Are there other experiments that would strengthen the paper further? How much would they improve it, and how difficult are they likely to be?

These experiments don't involve subduction, either continental or oceanic, so they are at a disadvantage compared with numerical models.

Our models involve subduction rollback and continental subduction roll-forward. They do not include the simple-shear down-dip subduction component, as explained in the Methods section. Besides, there are many numerical geodynamic models on the Himalaya and Tibet that do not involve subduction, e.g. England and Houseman [JGR 1989], Royden et al. [Science 1997], Copley et al. [Nature 2011], so we think the comment from reviewer 4 is not justified.

It appears to us that reviewer 4 is biased towards numerical modelling and against laboratory experiments. We think that both laboratory and numerical modelling are useful tools to understand the tectonics and geodynamics of tectonic plates and the mantle. Laboratory modelling and numerical modelling complement each other and have their own advantages and drawbacks. This is why my co-authors and I use both analogue modelling [e.g. Rosas et al., 2002, 2015; Schellart et al., 2003; Schellart, 2004, 2005, 2008, 2009, 2010; Duarte et al., 2013; Chen et al., 2016, 2017; Strak and Schellart, 2014, 2018] and numerical modelling [e.g. Schellart et al., 2007, 2010; Schellart, 2017; Rosas et al., 2012; Schellart and Moresi, 2013] to investigate geodynamic processes. For the current study we have chosen analogue models over numerical models, as they have a number of advantages compared to numerical models:

- The analogue models are inherently four-dimensional (3D space + time). Many numerical models are set up in 2D space.

- The analogue models can be conducted at a massive scale (continental lithosphere of 8000 km by 8800 km), while still retaining enough resolution at small scale to simulate shear localization and brittle faulting in continental lithosphere. So far, no numerical geodynamic model has been able to reproduce this.

- The models have a free top surface such that they can produce mountain topography and morphology that are comparable to that in the Tibet-Himalaya region, as well as basin bathymetry and morphology that are comparable to the marginal basins in East and Southeast Asia.

In any case, in the Methods section we now justify our choice for analogue modelling and how this provides an advantage over other geodynamic modelling methods.

Reviewer 4 only mentions two geodynamic modelling studies in his/her review, namely Capitanio et al. [2010] and Capitanio et al. [2015]. Capitanio et al. [2010] presented numerical models of Tethyan-Indian subduction which (1) had a two-dimensional spatial set-up and (2) did not include an overriding plate. The focus of the study of Capitanio et al. [2010] is on how continental subduction of the Indian lithosphere is possible. This is entirely different from our study, which focuses on continental deformation of the East Asian and Southeast Asian lithosphere. Capitanio et al. [2010] could not study this, because their models did not include an overriding Asian lithosphere.

The study of Capitanio et al. [2015] does have some relevance to our study, as their models had a 3D spatial set-up and involved an overriding plate. However, as noted above, their focus was on a much smaller area and excluded the entire Pacific subduction margin. Nevertheless, we now incorporate this work in the introduction part of the manuscript.

•Are the claims appropriately discussed in the context of previous literature?

No, see above. There are too many points where the model does not match the geology.

We have the impression that Reviewer 4 is not up to date with the recent literature that has been published on the extensional deformation in East and Southeast Asia, as shown in our Fig. 1a. The massive areal extent of Cenozoic extensional deformation in East and Southeast Asia is reproduced, for the first time, in our geodynamic models. This has never been

reproduced before in either experimental or numerical geodynamic models. In addition, our best-fitting models are able to reproduce the large-scale displacement patterns and mantle flow field patterns as implied by geodetic studies (Fig. 5) and seismic anisotropy studies (Fig. 1b,c, 7), and our best fitting models also agree with recent reconstructions on the amount of eastward continental extrusion and Indian indentation (Fig. 6). Finally, our models are the first to indicate that the Pacific subduction margin is responsible for causing east-west extension in Tibet.

To accommodate the comment from the reviewer, we now describe in a new paragraph how the results of our best fitting models agree with the first order Cenozoic geological, structural, tectonic, and morphological observations in East and Southeast Asia (also to accommodate a comment from reviewer 2).

•If the manuscript is unacceptable in its present form, does the study seem sufficiently promising that the authors should be encouraged to consider a resubmission in the future?

No, unfortunately. The models give a partial simulation of the upper crustal fault patterns across Asia, but it is too ambitious to claim that they shed light on the overall tectonics and dynamics.

See all of our responses to Reviewer 4 above. Our models are the first to reproduce widespread extension in East and Southeast Asia and link this with rollback of the Pacific and Sunda subduction margins. This has never been demonstrated or simulated before. We think this is a major break-through in the geosciences.

REVIEWERS' COMMENTS:

Reviewer #2 (Remarks to the Author):

Review Report on revised version entitled "Pacific subduction control on Tibetan extension and eastward extrusion tectonics and Asian continental deformation – by Schellart et al."

I am very happy and completely agree with all corrections made in the revised version of this manuscript. It is good to see that authors have also incorporated a new section to discuss the implication of model results for explaining spatial distribution of deformation structure in the Asian continent. In addition, authors also highlighted the importance of scale of observation for having a few discrepancies between the model results and natural observations. I strongly want this manuscript to publish soon.

I have a few minor comments on the newly added part of the Introduction section. Well, my suggestions are only to increase the readability of the manuscript.

Minor comments:

- (1) In the title I suggest to replace the 1st 'and' by ",".
- (2) In the abstract, my suggestion to include "Pacific" before rollback in line no. 25.
- (3) I think paragraph 2 and 3 in the introduction should be merged to a single paragraph.
- (4) I tried to reorganize the newly added last two paragraph of the Introduction, as shown in the annotated PDF.

Finally, I believe this work will open a new outlook for understanding the deformation in the Asian continent on a large scale. This paper will definitely attract readers from all groups of geosciences working in the field of collisional tectonics.

I strongly recommend the paper for publication at the earliest in Nature Communication.

Reviewer #4 (Remarks to the Author):

The revised version is better because it gives credit to previous work that proposed subduction rollback as a driver for east Asian tectonics in the Cenozoic. This improvement needs to be extended to the abstract, which still neglects to mention that the rollback model is not new, and still gives the misleading impression that this paper is the first work to propose the idea.

The authors were quite right to talk of the "prevailing model", but it was wrong to ignore the other papers that proposed rollback. Some of these papers have 100s of citations themselves – the Royden et al paper has >950, Watson et al has ~400. Other important references: An Yin (2010) Cenozoic tectonic evolution of Asia: A preliminary synthesis – discusses rollback (and has ~500 citations as we're counting). Ren et al (2002) also note rollback (see Figure 13) – a ~1000 citation paper.

As far back as 2004, Liu et al (Tectonophysics) were able to summarise the literature as "Many studies have linked Cenozoic rifting and volcanism in eastern China to the rollback of the subducting Pacific plate away from the Eurasian continent", and provide yet more references. In any case, what does it matter if one view is prevailing or not? The key thing is that the rollback model is not new, as implied by the way the authors set out the first version of the paper. This version is better – but as noted, the abstract should be improved.

This issue does not affect the modelling work in the paper; as clearly stated by the authors and noted in the original review, no one else has attempted analog modelling on this scale before. The point for Nature Communications is whether the lack of novelty in the rollback concept should prevent the journal from taking the paper.

Regarding the Cenozoic compressional deformation in the east Asian basins, the points about the Pliocene-Quaternary compressional deformation are well taken and well summarised, but there is earlier compressional deformation. See Daly et al (1991, MPG) for Indonesia and Su et al (2011, AAPG Bulletin) for a nice example from NE China. Much of this deformation is Oligocene – Early Miocene in age, and so predates the Tibetan rifting and Pliocene-Quaternary deformation discussed in the next section.

Re. right-lateral shear: see papers on the Cenozoic evolution of the Tan-Lu Fault, e.g. Hsiao et al (2004; AAPG Bulletin). The right-lateral transtensional basin development down the east side of China is well-known. See Suo et al (2015) "Continental margin basins in East Asia..." See An Yin (2010), again, and Allen et al (1998; Geol Soc London Special). This right-lateral oblique extension is a really important part of the story of Cenozoic east Asian tectonics - especially for hydrocarbons.

How much do the models need to reproduce every aspect of Asian deformation? Obviously they don't; the overall scope and patterns produced are impressive, and the authors are right that previous work involving lab models was not on this scale, whilst numerical models focussed on smaller parts of the system, or ignored certain aspects. Just be open about how much of the observed deformation is captured in the models and how much is not. Many parts of the paper do this well already. If it captures the points noted in the two paragraphs, above, it will be even better.

Response to reviewers

Please find our responses below (in Arial blue) to the comments from the reviewers (in Times New Roman black).

Reviewers' comments:

Reviewer 2:

Reviewer #2 (Remarks to the Author):

Review Report on revised version entitled “Pacific subduction control on Tibetan extension and eastward extrusion tectonics and Asian continental deformation – by Schellart et al.”

I am very happy and completely agree with all corrections made in the revised version of this manuscript. It is good to see that authors have also incorporated a new section to discuss the implication of model results for explaining spatial distribution of deformation structure in the Asian continent. In addition, authors also highlighted the importance of scale of observation for having a few discrepancies between the model results and natural observations. I strongly want this manuscript to publish soon.

I have a few minor comments on the newly added part of the Introduction section. Well, my suggestions are only to increase the readability of the manuscript.

Minor comments:

(1) In the title I suggest to replace the 1st ‘and’ by “,”.

Unfortunately, *Nature Communications* does not allow punctuation in the title. In any case, we have removed one of the “and” in the title and have rearranged it a little such that there is no need to have two “and” in the title. We think the title now reads much better.

(2) In the abstract, my suggestion to include “Pacific” before rollback in line no. 25.

We have added Pacific (and Sunda for completeness).

(3) I think paragraph 2 and 3 in the introduction should be merged to a single paragraph.

We have merged the two paragraphs.

(4) I tried to reorganize the newly added last two paragraph of the Introduction, as shown in the annotated PDF.

We have accommodated a number of suggestions proposed by reviewer 2 to improve readability. We have not accommodated all deletions proposed by reviewer 2, because *Nature Communications* guidelines require the last paragraph to contain a brief summary of the results and conclusions.

Finally, I believe this work will open a new outlook for understanding the deformation in the Asian continent on a large scale. This paper will definitely attract readers from all groups of geosciences working in the field of collisional tectonics.

I strongly recommend the paper for publication at the earliest in Nature Communication.

Reviewer 4:

Reviewer #4 (Remarks to the Author):

The revised version is better because it gives credit to previous work that proposed subduction rollback as a driver for east Asian tectonics in the Cenozoic. This improvement needs to be extended to the abstract, which still neglects to mention that the rollback model is not new, and still gives the misleading impression that this paper is the first work to propose the idea.

We do not agree that the abstract (nor the original manuscript) gives such a misleading impression. Nevertheless, we have modified the abstract (while staying within the 150-word limit), which removes any potential suggestion from our side that we are the first to propose a conceptual model describing the role of rollback in driving Cenozoic Asian deformation.

The authors were quite right to talk of the “prevailing model”, but it was wrong to ignore the other papers that proposed rollback. Some of these papers have 100s of citations themselves – the Royden et al paper has >950, Watson et al has ~400. Other important references: An Yin (2010) Cenozoic tectonic evolution of Asia: A preliminary synthesis – discusses rollback (and has ~500 citations as we’re counting). Ren et al (2002) also note rollback (see Figure 13) – a ~1000 citation paper.

We do not agree that we ignored other papers that proposed rollback. In our original version we included the works of Royden et al. [2008] and Ren et al. [2002], and in our first revision we also included Watson et al. [1987], following the suggestion of reviewer 4. Considering that *Nature Communications* has a limit on number of references, we cannot include an exhaustive list of references in our manuscript.

As far back as 2004, Liu et al (Tectonophysics) were able to summarise the literature as “Many studies have linked Cenozoic rifting and volcanism in eastern China to the rollback of the subducting Pacific plate away from the Eurasian continent”, and provide yet more references. In any case, what does it matter if one view is prevailing or not? The key thing is that the rollback model is not new, as implied by the way the authors set out the first version of the paper. This version is better – but as noted, the abstract should be improved.

We did not claim that our (conceptual) rollback model was new. Indeed, earlier works have proposed that local rollback has resulted in local backarc extension, e.g. Viallon et al. [1986] for the Okinawa Trough, Van Horne et al. [2017] and other scientists for the Japan Sea, Ren et al. [2002] for East China, Schellart et al. [2003] for the Kuril Basin and Sea of Okhotsk, Spakman and Hall [2010] for the Banda Sea. However, as far as we are aware, there are three earlier works [Fournier et al., 2004; Schellart and Lister, 2005; Royden et al., 2008] that have proposed rollback and extension on a much larger scale, i.e. ~eastward rollback of the western Pacific subduction zones and ~southward rollback of the Sunda subduction zone. These are the works that we already discussed in our original manuscript. What we do claim is that our modelling approach, with active rollback, is new. Our models allow us, for the first time, to test the hypothesis that rollback of the Pacific and Sunda subduction margins played an active role in Cenozoic extensional and strike-slip deformation in Central, East and Southeast Asia.

This issue does not affect the modelling work in the paper; as clearly stated by the authors and noted in the original review, no one else has attempted analog modelling on this scale before. The point for *Nature Communications* is whether the lack of novelty in the rollback concept should prevent the journal from taking the paper.

Geodynamic models allow the testing of conjectures and conceptual models, by probing the kinematic and dynamic robustness and physical viability of proposed ideas. Scientific novelty arises from the empirical confirmation (or rejection) of such different possibilities of explanations. To our knowledge, we are the first to investigate the role of Pacific and Sunda rollback on Asian deformation, in combination with India indentation, using geodynamic models.

Regarding the Cenozoic compressional deformation in the east Asian basins, the points about the Pliocene-Quaternary compressional deformation are well taken and well summarised, but there is earlier compressional deformation. See Daly et al (1991, MPG) for Indonesia and Su et al (2011, AAPG Bulletin) for a nice example from NE China. Much of this deformation is Oligocene – Early Miocene in age, and so predates the Tibetan rifting and Pliocene-Quaternary deformation discussed in the next section.

The implied major compressional event in the Sumatra region proposed by Daly [1991] with subduction polarity reversal in the late Oligocene and then again another polarity reversal

back to the present-day setting is not shown in more recent reconstructions that use a much larger data base [e.g. Hall, 2012]. A ~1 Myr phase of compression in the latest Oligocene in southern Sumatra is shown in a detailed study of basins in Southeast Asia by Doust and Noble [2008], but such a local, short-lived phase of overriding plate compression and shortening can be explained, for example, by subduction of an aseismic ridge or small oceanic plateau [e.g. Flórez-Rodríguez et al., 2019]. We can apply a similar geodynamic explanation to the short-lived late Oligocene inversion phase of the Chezhen basin, a sub-basin of the Bohai basin in NE China, as documented by Su et al. [2011]. In any case, we have added part of the text from above to the manuscript (Discussion section) and added the references Su et al. [2011], Flórez-Rodríguez et al. [2019] and Doust and Noble [2008] to accommodate the comment from the reviewer.

Re. right-lateral shear: see papers on the Cenozoic evolution of the Tan-Lu Fault, e.g. Hsaio et al (2004; AAPG Bulletin). The right-lateral transtensional basin development down the east side of China is well-known. See Suo et al (2015) “Continental margin basins in East Asia...” See An Yin (2010), again, and Allen et al (1998; Geol Soc London Special). This right-lateral oblique extension is a really important part of the story of Cenozoic east Asian tectonics - especially for hydrocarbons. We have incorporated a brief discussion in the text on the Tan-Lu fault, a large fault in East China that is shown (and is now also labelled) in our Fig. 1a, as well as the right-lateral transtension in the Bohai Bay Basin west of the fault. Although the Tan-Lu fault is hundreds of km long, detailed studies indicate that the shearing along the fault did not play a major role in Paleogene extension in the region [Hsaio et al., 2004], that the Cenozoic dextral offset is only of the order of 21 km and might have resulted from the oblique subduction of the Pacific plate, and that the fault accommodated oblique extension (dextral transtension) [Huang et al., 2015; Allen et al., 1998]. In any case, we have added part of the text from above to the manuscript in the Discussion section and added the references Allen et al. [1998] and Huang et al. [2015] to accommodate the comment from the reviewer.

How much do the models need to reproduce every aspect of Asian deformation? Obviously they don't; the overall scope and patterns produced are impressive, and the authors are right that previous work involving lab models was not on this scale, whilst numerical models focussed on smaller parts of the system, or ignored certain aspects. Just be open about how much of the observed deformation is captured in the models and how much is not. Many parts of the paper do this well already. If it captures the points noted in the two paragraphs, above, it will be even better. We have accommodated the comments presented in the reviewer's two paragraphs above into the revised manuscript.